# Observation-constrained estimates of the global ocean carbon sink from Earth System Models

Jens Terhaar[1,2], Thomas L. Frölicher[1,2], Fortunat Joos[1,2]

[1]Climate and Environmental Physics, Physics Institute, University of Bern, Switzerland
[2]Oeschger Centre for Climate Change Research, University of Bern, Switzerland

*Correspondence to*: Jens Terhaar (jens.terhaar@unibe.ch)

**Abstract.** The ocean slows global warming by currently taking up around one quarter of all human-made $CO_2$ emissions. However, estimates of the ocean anthropogenic carbon uptake vary across various observation-based and model-based approaches. Here, we show that the global ocean anthropogenic carbon sink simulated by Earth System Models can be

constrained by two physical parameters, the present-day sea surface salinity in the subtropical-polar frontal zone in the Southern Ocean and the strength of the Atlantic Meridional Overturning Circulation, and one biogeochemical parameter, the Revelle factor of the global surface ocean. The Revelle factor quantifies the chemical capacity of seawater to take up carbon for a given increase in atmospheric $CO_2$. By exploiting this three-dimensional emergent constraint with observations, we provide a new model- and observation-based estimate of the past, present, and future global ocean anthropogenic carbon sink

and show that the ocean carbon sink is 9-11% larger than previously estimated. Furthermore, the constraint reduces uncertainties of the past and present global ocean anthropogenic carbon sink by 42-59% and the future sink by 32-62% depending on the scenario, allowing for a better understanding of the global carbon cycle and better targeted climate and ocean policies. Our constrained results are in good agreement with the air-sea $C_{ant}$ estimates over the last three decades based on observations of the $CO_2$ partial pressure at the ocean surface in the Global Carbon Budget 2021, and suggest that existing

hindcast ocean-only model simulations underestimate the global ocean anthropogenic carbon sink. The here identified key parameters for the ocean carbon sink should be quantified when presenting simulated ocean anthropogenic carbon uptake as in the Global Carbon Budget and be used to adjust these simulated estimates if necessary. The larger ocean sink results in enhanced ocean acidification over the 21st century, which further threatens marine ecosystems by reducing the water volume that is projected to be undersaturated towards aragonite by around 3.7-7.4 million $km^3$ more than originally projected.

## 1 Introduction

The emissions of anthropogenic $CO_2$ ($C_{ant}$) since the beginning of the industrialization through fossil-fuel burning, cement production and land-use change have altered the global carbon cycle and climate (Friedlingstein et al., 2022). Around 40% of the additional carbon since 1850 has accumulated in the atmosphere, where it represents the main anthropogenic greenhouse

gas (IPCC, 2021). More than half of the emitted $C_{ant}$ has been taken up by the land biosphere (~30%) and the ocean (~25%)

(Friedlingstein et al., 2022). The remaining ~5% are the budget imbalance, a mismatch between carbon emissions and sink

estimates which cannot be explained yet (Friedlingstein et al., 2022). By taking up each around a quarter of the $C_{ant}$ emissions,

the land biosphere and ocean sinks slow down global warming and climate change.

The ocean $C_{ant}$ sink is defined here as a combination of the uptake of newly emitted carbon and the change in the natural carbon

inventory in the ocean due to changes in temperatures, winds, and the freshwater cycle caused by climate change (Joos et al.,

1999; Frölicher and Joos, 2010; McNeil and Matear, 2013). The uptake rate of $C_{ant}$ on sub-millennial timescales is mainly

determined by the ocean circulation and carbonate chemistry and only partly by biology (Sarmiento et al., 1998; Joos et al.,

1999; Caldeira and Duffy, 2000; Sabine et al., 2004), despite the overall importance of marine biology for natural carbon

fluxes (Falkowski et al., 1998; Steinacher et al., 2010). The rate limiting process of $C_{ant}$ uptake is the circulation that transports

surface waters with high $C_{ant}$ concentrations into the deeper ocean and allows waters with low or no $C_{ant}$ concentrations to

upwell back to the ocean surface. The largest part of this ocean upwelling occurs in the Southern Ocean where strong westerlies

drive northward Ekman transport of surface waters, which are then replaced by older, deeper water masses (Marshall and

Speer, 2012; Talley, 2013; Morrison et al., 2015). These predominantly northward flowing waters take up $C_{ant}$ from the

atmosphere and are eventually transferred to mode and intermediate waters that sink back into the ocean interior (Marshall and

Speer, 2012; Talley, 2013). This overturning makes the Southern Ocean the largest marine $C_{ant}$ sink (~40% of global ocean

$C_{ant}$ uptake) (Caldeira and Duffy, 2000; Mikaloff Fletcher et al., 2006; Frölicher et al., 2015; Terhaar et al., 2021b). Another

region of large uptake rates is the North Atlantic (Caldeira and Duffy, 2000; Mikaloff Fletcher et al., 2006), where the Atlantic

Meridional Overturning Circulation (AMOC) transports surface waters with high $C_{ant}$ (Pérez et al., 2013) and subsurface waters

with low $C_{ant}$ concentrations northward (Ridge and McKinley, 2020). The subsurface waters outcrop in the subpolar North

Atlantic where they take up $C_{ant}$ from the atmosphere (Ridge and McKinley, 2020). These high $C_{ant}$ waters are then ventilated

by the AMOC into the deep ocean where the $C_{ant}$ is efficiently stored (Joos et al., 1999; Winton et al., 2013).

While the circulation determines the volume that is transported into the deeper ocean, the Revelle factor (Revelle and Suess, 1957; Sabine et al., 2004) determines the concentration of $C_{ant}$ in these water masses. The Revelle factor describes the

biogeochemical capacity of the ocean to take up $C_{ant}$. This biogeochemical capacity is strongly dependent on the amount of carbonate ions in the ocean that react with $CO_2$ and $H_2O$ to form bicarbonate ions (Egleston et al., 2010; Goodwin et al., 2009; Revelle and Suess, 1957). The more $CO_2$ is transferred via this reaction to bicarbonate ions, the more can be taken up again from the atmosphere. The available amount of carbonate ions for this reaction depends sensitively on the difference between ocean alkalinity and dissolved inorganic carbon ($C_T$) (Figure A2) (Egleston et al., 2010; Goodwin et al., 2009; Revelle and

Suess, 1957), highlighting the importance of alkalinity for the global ocean carbon uptake (Middelburg et al., 2020). As the buffer factor influences the $C_{ant}$ uptake, it also exerts a strong control on the transient climate response, i.e., the warming per cumulative $CO_2$ emissions (Katavouta et al., 2018; Rodgers et al., 2020).

In addition to slowing global warming, the $C_{ant}$ uptake by the ocean also causes ocean acidification (Orr et al., 2005; Gattuso

and Hansson, 2011; Kwiatkowski et al., 2020), i.e., a decline in ocean pH and carbonate ion concentrations. The decline in carbonate ion concentrations has negative effects on the growth and survival of many marine species, especially on calcifying organisms whose shells and skeletons are made up of calcium carbonate minerals (Orr et al., 2005; Fabry et al., 2008; Kroeker et al., 2010, 2013; Doney et al., 2020). Calcium carbonate minerals in the ocean exists mainly in its metastable forms of aragonite and high-magnesium calcite and its more stable form calcite. The stability of calcium carbonate minerals is described

by their saturation states ($\Omega$), which describe the product of the concentrations of calcium ($[Ca^{2+}]$) and carbonate ions ($[CO_3^{2-}]$) divided by their product in equilibrium. Reductions of saturation states of aragonite ($\Omega_{arag}$) and calcite ($\Omega_{calc}$) have shown to negatively impact organisms and ecosystems (Langdon and Atkinson, 2005; Kroeker et al., 2010; Bednaršek et al., 2014; Albright et al., 2016). Once, saturation states drop below one, the water is undersaturated and actively corrosive towards the respective mineral form.


Accurately quantifying the ocean anthropogenic carbon sink is thus of crucial importance for understanding and quantifying the carbon cycle, global warming and climate change, as well as ocean acidification. A better knowledge of the size of the

historical and future ocean carbon sink and reduced uncertainties will hence not only lead to an improved understanding of the overall carbon cycle and global climate change (IPCC, 2021), but also allow targeted climate and ocean policies (IPCC, 2022).

One of the key tools to assess the past, present, and future ocean carbon sink are Earth System Models (ESMs). However, the simulated ocean $C_{ant}$ sink varies across the different ESMs (Frölicher et al., 2015; Wang et al., 2016; Bronselaer et al., 2017; Terhaar et al., 2021b) and the model differences grow over time, i.e., ESMs that simulate a small ocean $C_{ant}$ uptake over the last decades also simulate a small uptake over the 21$^{st}$ century (Figure 1b) (Wang et al., 2016). Therefore, a better knowledge of the ocean $C_{ant}$ sink in the last decades would be one possibility to reduce uncertainties in the simulated ocean carbon from

1850 to 2100.

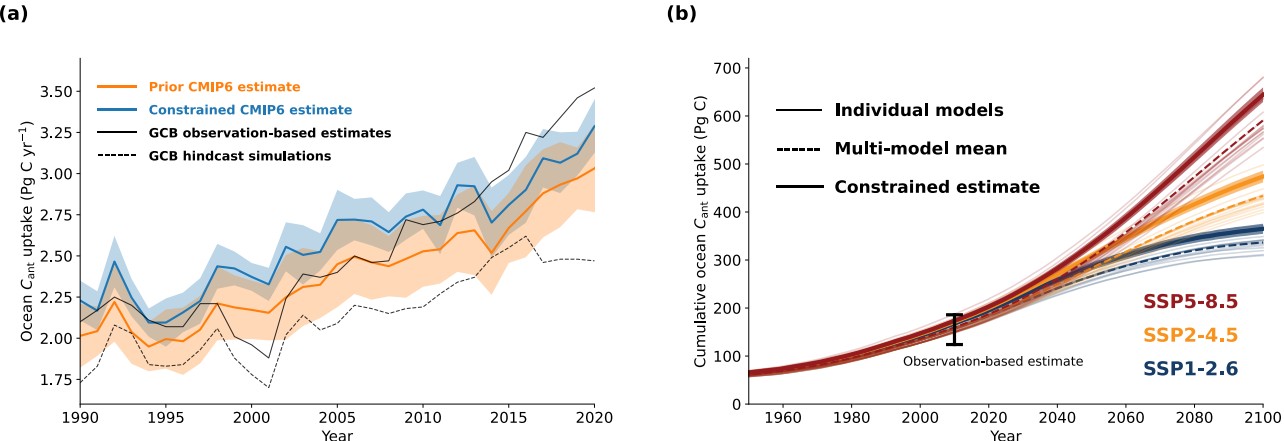

**Figure 1. Simulated ocean anthropogenic carbon uptake from Earth System Models. (a)** Simulated annual mean air-sea $C_{ant}$ fluxes from 17 CMIP6 Earth System Models from 1990 to 2020 before (orange line) and after the constraint is applied (blue line). After 2014, results from SSP5-8.5 were chosen as this is the only SSP for which each model provided results and differences in atmospheric $CO_2$ mixing
ratios in SSP5-8.5 (Meinshausen et al., 2020) are small compared to observations until 2020 (maximum difference of 2.5 ppm in 2020) (Trends in Atmospheric Carbon Dioxide (NOAA/GML)). In addition, mean air-sea $C_{ant}$ fluxes based on multiple observation-based estimates (black solid line) and hindcast simulations (black dashed line) from the Global Carbon Budget 2021 (Friedlingstein et al., 2022) are shown. For readability, the uncertainties of these estimates (on average 0.24 Pg C yr$^{-1}$ for observation-based estimates and 0.28 Pg C yr$^{-1}$ for hindcast simulations) are not shown in the figure. **(b)** Simulated cumulative ocean $C_{ant}$ uptake since 1765 for the historic period until 2014 (17 ESMs)
and for the future from 2015 to 2100 under SSP1-2.6 (blue, 14 ESMs), SSP2-4.5 (orange, 16 ESMs), and SSP5-8.5 (red, 17 ESMs). Thin lines show the results from each individual ESM, the dashed lines the multi-model mean, the solid lines the constrained estimate, and the shading the uncertainty around the constrained estimate. Furthermore, the observation-based ocean $C_{ant}$ inventory estimate in 2010 from Khatiwala et al. (2013) is shown. As ESM simulations in CMIP6 start in 1850, the air-sea $C_{ant}$ fluxes were corrected upwards for the late starting date in the constrained estimate following Bronselaer et al. (2017) (see Appendix A.1).

## 2 Quantifying the past ocean anthropogenic carbon sink with observations and hindcast simulations and existing uncertainties

The large background concentration of $C_T$ in the ocean and the vast ocean volume make it difficult to directly observe the relatively small anthropogenic perturbations in the ocean interior. Therefore, different methods have been developed to estimate the accumulation of anthropogenic carbon ($C_{ant}$) in the ocean (Khatiwala et al., 2013), such as the $\Delta C^*$ method (Gruber et al., 1996; Sabine et al., 2004) or the Transient Time Distribution method (Hall et al., 2002) based on observations of inert tracers, like CFCs. These estimates result in an estimated ocean $C_{ant}$ inventory in 2010 of 155±31 Pg C (Khatiwala et al., 2013) (Figure 1b, Table 1), but do not or only partly include climate-driven changes in $C_T$.

Further development of the $\Delta C^*$ method into the eMLR(C*) method (Clement and Gruber, 2018) and more observations through new techniques, such as (Bio-)ARGO-floats (Claustre et al., 2020), and more research cruises (Lauvset et al., 2021) allowed to quantify the increase in marine $C_{ant}$ on shorter timescales and with reduced uncertainty. The increase in $C_{ant}$ from 1994 to 2007 by the eMLR(C*) method is 34±4 Pg C (12% uncertainty, Table 1) (Gruber et al., 2019a), again not accounting for potential climate-driven changes in $C_T$. In addition to interior $C_{ant}$ estimates, surface ocean observations of the partial pressure of $CO_2$ ($pCO_2$) and new statistical methods, such as neural networks (Landschützer et al., 2016), have led to a variety of observation-based estimates of the air-sea $CO_2$ flux (Rödenbeck et al., 2014; Zeng et al., 2014; Landschützer et al., 2016; Gregor et al., 2019; Watson et al., 2020; Iida et al., 2021; Gregor and Gruber, 2021; Chau et al., 2022). When subtracting the pre-industrial outflux of $CO_2$ due to riverine carbon fluxes (Sarmiento and Sundquist, 1992; Aumont et al., 2001; Jacobson et al., 2007; Resplandy et al., 2018; Lacroix et al., 2020; Regnier et al., 2022) from these air-sea $CO_2$ flux estimates, the global ocean $C_{ant}$ uptake can be derived (Friedlingstein et al., 2022), resulting in an estimated ocean $C_{ant}$ uptake from 1994 to 2007 of 29±4 Pg C (14% uncertainty, Table 1).

The difference of 5 Pg C between the interior and surface ocean mean estimates was attributed to outgassing of ocean $CO_2$ caused by a changing climate and climate variability (Gruber et al., 2019a). However, simulations from ESMs of the sixth phase of the Coupled Model Intercomparison Project (CMIP6) estimate the climate-driven and externally forced climate variability-drive air-sea $CO_2$ flux from 1994 to 2007 to be only -1.6±0.5 Pg C (Table A3). When averaging over an ensemble

of ESMs, forced variability (e.g., due to the volcanic eruptions or varying emissions of $CO_2$ and other radiative agents) is still preserved. However, unforced interannual-to-decadal variability is largely removed when averaging over an ensemble of ESMs. Although comparisons suggest that the ocean $C_{ant}$ uptake was low compared to atmospheric $CO_2$ in the 1990s and high in the 2000s (Rödenbeck et al., 2013, 2022), a comparison of different $C_{ant}$ uptake estimates for different decadal-scale periods does not reveal any clear variability-related deviation for the 1994-2007 period (IPCC, WGI, Chapter 5, Figure 5.8 (Canadell et al., 2021)). Overall, uncertainties remain at present too large for any quantitative conclusions, but it seems unlikely that unforced variability causes an air-sea $CO_2$ flux of -3.4 Pg C (difference between -5 Pg C from Gruber et al. (2019a) and -1.6 Pg C from ESMs), twice as large as the simulated flux from forced variability and climate change. It hence remains a challenge to derive the total ocean $C_{ant}$ sink from interior estimates that do not account for climate-driven changes in $C_T$.

An alternative way of estimating the strength of the ocean carbon sink is the use of global ocean biogeochemical models forced with atmospheric reanalysis data (Sarmiento et al., 1992; Friedlingstein et al., 2022). From 1994 to 2007, the ocean biogeochemical hindcast models that participated in the Global Carbon Budget 2021 (Friedlingstein et al., 2022) simulate a $C_{ant}$ uptake of 26±3 Pg C (Table 1). This estimate is 3 Pg C below the surface observation-based estimate and the difference increases further after 2010 (Figure 1a). Compared to the interior ocean $C_{ant}$ estimate, the simulated uptake by these hindcast models is 3-6 Pg C (10-19%) smaller depending on the correction term that is used for climate change induced outgassing of natural $CO_2$. Such differences between observation-based and simulated ocean $C_{ant}$ uptake could be explained regionally by systematic biases in models (Goris et al., 2018; Terhaar et al., 2020a, 2021a, b), as well as data sparsity (Bushinsky et al., 2019; Gloege et al., 2021).

Overall, the difference between ocean hindcast models, observation-based $CO_2$ flux estimates, and interior ocean $C_{ant}$ estimates as well as the uncertainties in the climate-driven change in $C_T$ and pre-industrial outgassing indicate that uncertainties of the ocean $C_{ant}$ sink over the last decades remain substantial. The uncertainty of the $C_{ant}$ sink appears larger than the uncertainty typically given for an individual estimate of the $C_{ant}$ sink from a specific data product.

**Table 1. Global ocean air-sea $C_{ant}$ flux estimates based on 17 ESMs from CMIP6 before and after starting date corrected and constraint as well as previous estimates over different time periods. Prior uncertainty is the multi-model standard deviation. The uncertainty of the starting date corrected values also includes the uncertainty from that correction. The constrained uncertainty is a combination of the starting date correction, the multi-model standard deviation after the constraint is applied, and the uncertainty from the correction itself (see section 3.1 and appendix A.1). Uncertainties from the decadal variability on shorter timescales, e.g., for 1994-2007, are not included. The star indicates estimates that do not account for climate-driven changes in the ocean carbon sink.**

| Period | Cumulative air-sea $C_{ant}$ flux (Pg C) | | | | | |
|---|---|---|---|---|---|---|
| | CMIP6 | | | Global Carbon Budget 2021 (Friedlingstein et al., 2022) | Others | |
| | Prior | Starting date corrected | Constrained | observation-based / hindcast simulations | Estimate | Source |
| 1994-2007 | 26.8 ± 2.1 | 28.8 ± 2.2 | 31.5 ± 0.9 | 29 ± 4 / 26 ±3 | 34 ± 4* | (Gruber et al., 2019a) |
| 1990-2020 | 69.7 ± 5.1 | 74.4 ± 5.4 | 80.7 ± 2.5 | 81 ± 7 / 68 ± 8 | | |
| 1765-2010 | | 164 ± 12 | 177 ± 7 | | 155 ± 31* | (Khatiwala et al., 2013) |
| 1850-2014 | 138 ± 10 | 157 ± 12 | 171 ± 6 | 150 ± 30 | | |
| 1960-2020 | 106 ± 8 | 117 ± 9 | 128 ± 4 | 115 ± 25 | | |
| 1850-2020 | 154 ± 11 | 174 ± 13 | 189 ± 7 | 170 ± 35 | | |
| 2020-2100 (SSP1-2.6) | 150 ± 11 | 156 ± 11 | 173 ± 8 | | | |
| 2020-2100 (SSP2-4.5) | 244 ± 16 | 251 ± 17 | 277 ± 9 | | | |
| 2020-2100 (SSP5-8.5) | 399 ± 29 | 407 ± 30 | 445 ± 12 | | | |

## 3 Constraining the ocean anthropogenic carbon sink in Earth System Models

Another way to constrain the past, present and future global ocean anthropogenic carbon sink is the use of process-based emergent constraints (Orr, 2002) that identify a relationship across an ensemble of ESMs between a relatively uncertain variable, such as the $C_{ant}$ uptake in the Southern Ocean, and a variable that can be observed with a relatively small uncertainty, such as the sea surface salinity in the subtropical-polar frontal zone in the Southern Ocean. The identified relationship is then combined with observations, in this example the sea surface salinity, to better estimate the uncertain variable, here the $C_{ant}$ uptake in the Southern Ocean (Terhaar et al., 2021b). Such relationships must be explainable by an underlying mechanism (Hall et al., 2019), i.e., higher sea surface salinity in the frontal zone leads to denser sea surface waters and stronger mode and intermediate water formation, which enhances the transport of $C_{ant}$ from the ocean surface to the ocean interior and allows hence for more $C_{ant}$ uptake. In recent years, process-based emergent constraints (Orr, 2002; Matsumoto et al., 2004; Wenzel et al., 2014; Kwiatkowski et al., 2017; Goris et al., 2018; Eyring et al., 2019; Hall et al., 2019; Terhaar et al., 2020a, 2021a, b; Bourgeois et al., 2022) have successfully reduced uncertainties in simulated fluxes across ensembles of ESMs. In the ocean, for example, a bias towards too little $C_{ant}$ uptake was identified in the Southern Ocean (Terhaar et al., 2021b). Similarly, ESMs from CMIP5 were shown to underestimate the future uptake of $C_{ant}$ in the North Atlantic due to too little sequestration of $C_{ant}$ into the deeper ocean (Goris et al., 2018). However, the relatively uncertain observation-based estimates of $C_{ant}$ sequestration (see section above) did not allow to reduce uncertainties. Similarly, the $C_{ant}$ uptake in the tropical Pacific Ocean across ESMs could be reduced with observations of the local surface ocean carbonate ion concentrations (Vaittinada Ayar et al., 2022), which is anti-correlated to the Revelle factor. Despite a better understanding of the regional $C_{ant}$ uptake, uncertainties of the global ocean $C_{ant}$ sink have not been reduced yet.

Here, we identify a mechanistic constraint for the global ocean $C_{ant}$ sink across 17 ESMs from CMIP6 (Table A1). We demonstrate that a linear combination of three observable quantities, (1) the sea surface salinity in the subtropical-polar frontal zone in the Southern Ocean, (2) the strength of the AMOC at 26.5°N, and (3) the globally averaged surface ocean Revelle factor, can successfully predict the strength of the global ocean $C_{ant}$ sink across the CMIP6 ESMs ($r^2$ of 0.87 for the global ocean $C_{ant}$ uptake from 1994 to 2007). The sea surface salinity in the subtropical-polar frontal zone in the Southern Ocean and

185 the AMOC determine the strength of the two most important regions of mode, intermediate, and deep-water formation (Goris

et al., 2018, 2022; Terhaar et al., 2021b). In addition, the Revelle factor accounts for biases in the biogeochemical buffer

capacity of the ocean, i.e., the relative increase in ocean $C_T$ for a given relative increase in ocean $pCO_2$ (Revelle and Suess,

1957). As the Revelle factor quantifies relative increases in ocean $C_T$, the increase in surface ocean $C_{ant}$ depends on the Revelle

factor and the natural surface ocean $C_T$. Therefore, the Revelle factor in the ESMs was adjusted for model biases in natural

surface ocean $C_T$ (see Appendix A.1). Compared to observations, CMIP6 models represent the observation-based average

strength of the AMOC from 2004 to 2020 ($16.91 \pm 0.49$ Sv) (McCarthy et al., 2020) right but have a large inter-model spread

($16.91 \pm 3.00$ Sv), underestimate the observed inter-frontal sea surface salinity ($34.07 \pm 0.02$) and have a large inter-model

spread ($33.89 \pm 0.13$), and overestimate the surface-averaged Revelle factor that was derived by GLODAPv2 ($10.45 \pm 0.01$)

by 0.24 ($10.73 \pm 0.24$) with largest Revelle factor biases in the main $C_{ant}$ uptake regions (Figure 2). The underestimation of the

195 $C_T$-adjusted Revelle factor by the ESM ensemble is mainly due to a bias towards too small concentrations of surface ocean

carbonate ion concentrations (Sarmiento et al., 1995), caused by a too small difference of surface ocean alkalinity and $C_T$

(Figure A2).

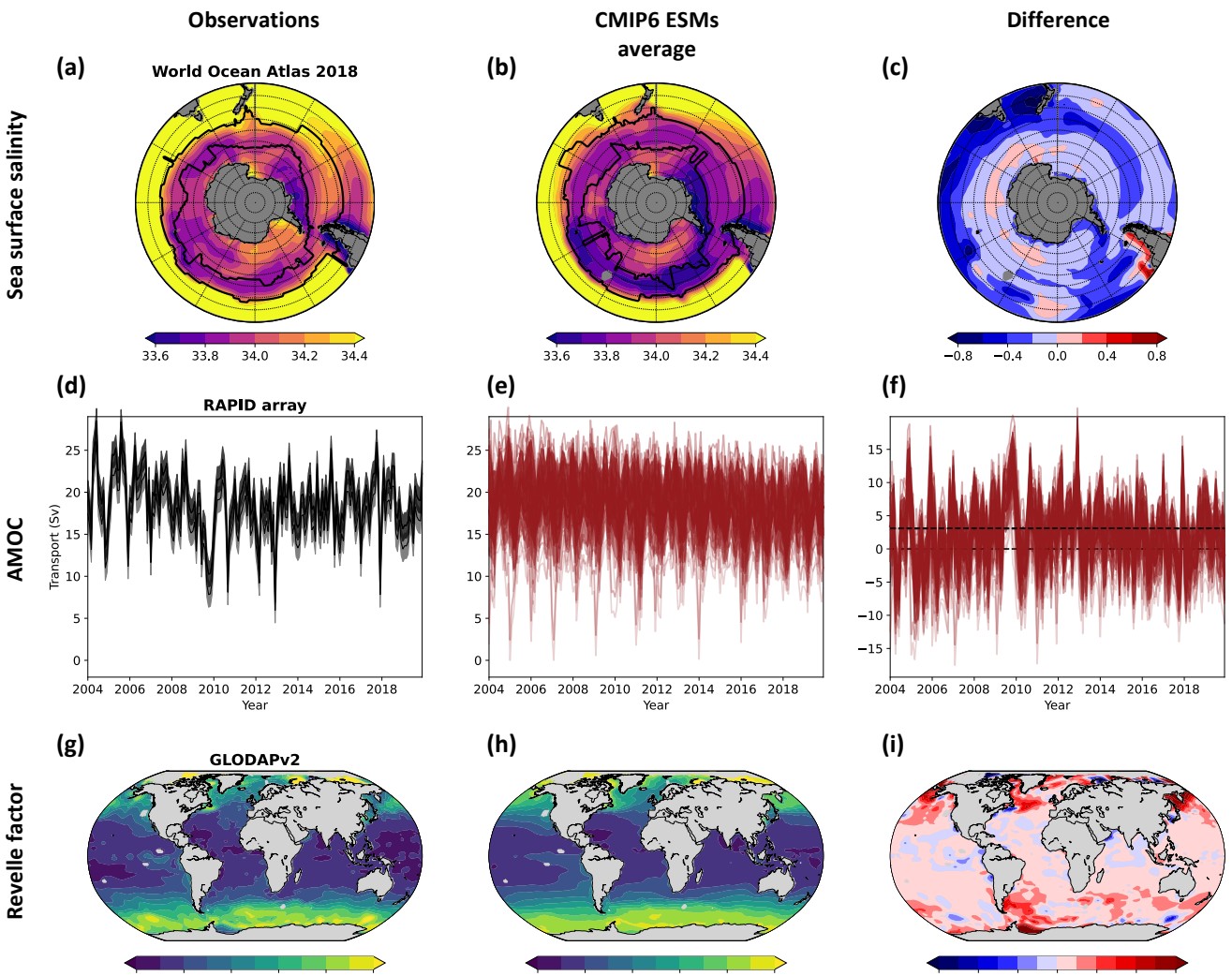

**Figure 2. Sea surface salinity in the Southern Ocean, the Atlantic Meridional Overturning Circulation, and the Revelle factor at the ocean surface from observations and Earth System Models.** Annual mean sea surface salinity from the **(a)** World Ocean Atlas 2018 (Zweng et al., 2018; Locarnini et al., 2018), **(b)** 17 Earth System Models from CMIP6 from 1995 to 2014, and **(c)** the difference between both. The black lines in **(a,b)** indicate the annual mean positions of the Polar and Subtropical Fronts. The strength of the monthly-averaged Atlantic Meridional Overturning Circulation, here defined as the maximum of the streamfunction at 26.5°N, from 2004 to 2020 as **(d)** observed by the RAPID array (McCarthy et al., 2020), **(e)** as simulated by 17 Earth System Models from CMIP6, and **(f)** the difference between both. Each model simulation is shown in **(e)** and **(f)** as a thin red line, the multi-model average is shown as a thick red line, and the multi-model standard deviation is shown as red shading. The annual mean sea surface Revelle factor calculated with *mocsy2.0* (Orr and Epitalon, 2015) from **(g)** gridded GLODAPv2 observations that are normalized to the year 2002 (Lauvset et al., 2016), from **(h)** output of 17 Earth System Model simulations from CMIP6 in 2002 and adjusted for biases in the surface ocean $C_T$ (see Appendix A.1), and **(i)** their difference.

### 3.1 Applying the constraint and uncertainty estimation

For the three-dimensional emergent constraint, multi-linear regression was used. First, it was assumed that the ocean $C_{ant}$ uptake for every model $M$ ($C_{ant}^M$) can be approximated by a linear combination of the inter-frontal sea surface salinity in the Southern Ocean in model $M$ ($SSS_{Southern\ Ocean}^M$), the AMOC strength in model $M$ ($AMOC^M$), and the globally-averaged surface ocean Revelle factor in model $M$ ($Revelle_{global}^M$):

$$C_{ant}^M = a * SSS_{Southern\ Ocean}^M + b * AMOC^M + c * Revelle_{global}^M + d + \varepsilon. \tag{1}$$

The parameters $a$, $b$, and $c$ are scaling parameters of the three predictor variables, $d$ is the y intercept, and $\varepsilon$ describes the residual between the predicted $C_{ant}$ flux by this multi-linear regression model and the simulated $C_{ant}$ uptake by model $M$. The free parameters $a$, $b$, $c$, and $d$ were fitted based on the simulated inter-frontal sea surface salinity in the Southern Ocean, AMOC, Revelle factor, and $C_{ant}$ uptake. The three predictors are not statistically correlated ($r^2 = 0.00$ for salinity and AMOC, $r^2 = 0.03$ for Revelle factor and AMOC, and $r^2 = 0.10$ for salinity and Revelle factor) and can hence be used in a multi-linear regression.

The constrained $C_{ant}$ flux is estimated by replacing the simulated inter-frontal sea surface salinity in the Southern Ocean, AMOC, and Revelle factor by the observed ones and by setting $\varepsilon$ to zero. As the Revelle factor describes the inverse of the ocean capacity to take up $C_{ant}$ from the atmosphere, equation (1) should in principal be used with $\frac{1}{Revelle_{global}^M}$. However, using $Revelle_{global}^M$ facilitates understanding and the presentation of the results and only introduces maximum errors of around 0.1% for the Revelle factor adjustment for the models that simulate the largest deviations from the observed Revelle factor. To estimate the uncertainty, all model results were first corrected for their biases in the three predictor variables, i.e., if a model has a salinity that is 0.2 smaller than the observed salinity, the simulated $C_{ant}$ uptake by this model is increased by $a * 0.2$. The same correction is made for the other two predictor variables (Figure 3). If the three predictor variables were predicting the $C_{ant}$ flux perfectly, the bias-corrected $C_{ant}$ uptake from all models would be the same. The remaining inter-model standard

deviation therefore represents the uncertainty from the multi-linear regression model due to other factors that influence the

ocean $C_{ant}$ uptake. The second part of the uncertainty originates from the uncertainty in the observations of the predictor

variables that influences the magnitude of the correction. This uncertainty ($\Delta C_{ant}^{obs}$) is calculated as follows:

$$\Delta C_{ant}^{obs} = \sqrt{(a * \Delta SSS_{Southern\ Ocean}^{obs})^2 + (b * \Delta AMOC^{obs})^2 + \left(c * \Delta Revelle_{global}^{obs}\right)^2}, \tag{2}$$

with $\Delta SSS_{Southern\ Ocean}^{obs}$, $\Delta AMOC^{obs}$, and $\Delta Revelle_{global}^{obs}$ being the uncertainty of the three observed predictor variables.

Eventually, the overall uncertainty of this constrained $C_{ant}$ flux is estimated as the square-root of the sum of the products of the

square of both uncertainties.

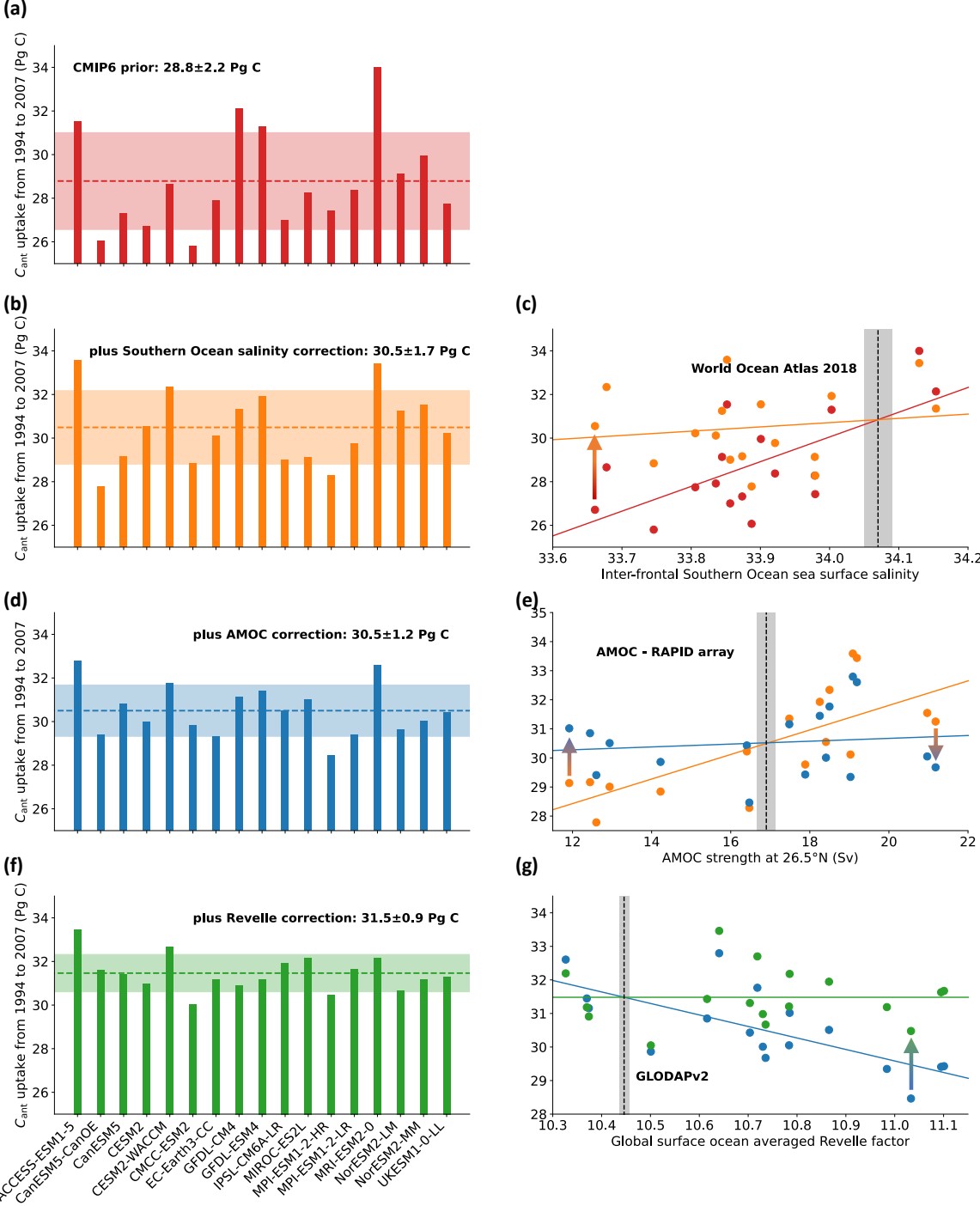

**Figure 3. Global ocean anthropogenic carbon simulated by Earth System Models from CMIP6 corrected for biases in sea surface salinity in the Southern Ocean, the Atlantic Meridional Overturning Circulation, and the Revelle factor. (a)** Global ocean

anthropogenic carbon ($C_{ant}$) uptake from 1994 to 2007 as simulated by 17 ESMs from CMIP6 and corrected for the late starting date
(Bronselaer et al., 2017). For each ESM, one ensemble member was used as the difference between ensemble members has been shown to
be small compared to the inter-model differences (Terhaar et al., 2020a, 2021b). In the years 1994 and 2007, only half of the annual $C_{ant}$
uptake was accounted for to make it comparable to interior ocean estimates that compare changes in $C_{ant}$ from mid 1994 to mid 2007 and not
from the start of 1994 to the end of 2007 (Gruber et al., 2019a). **(b)** $C_{ant}$ uptake after correcting the simulated $C_{ant}$ uptake from **(a)** for biases
in the Southern Ocean Sea surface salinity (Terhaar et al., 2021b) from **(c)**. The dots in **(c)** represent individual models before (red) and after
(orange) the sea surface salinity correction. **(d)** $C_{ant}$ uptake after correcting sea surface salinity corrected $C_{ant}$ uptake from **(b)** for biases in
the Atlantic Meridional Overturning Circulation from **(e)**. The dots in **(e)** represent individual models before (orange) and after (blue) the
Atlantic Meridional Overturning Circulation correction. **(f)** $C_{ant}$ uptake after correcting the sea surface salinity and Atlantic Meridional
Overturning Circulation corrected $C_{ant}$ uptake from **(d)** for biases in the global ocean surface Revelle factor from **(g)**. The dots in **(g)** represent
individual models before (blue) and after (green) the Revelle factor correction. The simulated Revelle factor by the ESMs was adjusted for
biases in the surface ocean $C_T$ (see Appendix A.1). The dashed coloured lines in **(a)**, **(b)**, **(d)**, **(f)** show the multi-model mean and the shading
shows the uncertainty, which is a combination of the multi-model standard deviation after correction and the uncertainty of the correction
factor due to the uncertainty of the observational constraint (see Appendix A.1). The dashed black lines in **(c)**, **(e)**, **(g)** show the observations
from the World Ocean Atlas 2018 (Zweng et al., 2018; Locarnini et al., 2018), the RAPID array (McCarthy et al., 2020), and GLODAPv2
(Lauvset et al., 2016) with their uncertainties as grey shading, the coloured lines show linear fits, and the arrows illustrate the correction for
individual models.

## 3.2 Exploiting the constraint with observations

By exploiting this multi-variable emergent constraint with observations, the simulated $C_{ant}$ uptake by ESMs from 1994 to 2007
increases from $28.8 \pm 2.2$ Pg C to $31.5 \pm 0.9$ Pg C (Figures 1 & 3, Tables 1 & A2). Biases in the Southern Ocean salinity are
responsible for around 60% of the bias in the global ocean $C_{ant}$ uptake in the CMIP6 models while the bias in the Revelle factor
explains the remaining 40% (Figure 3). The AMOC, whose multi-model mean in ESMs is similar to observations, does not
change the central $C_{ant}$ uptake estimate but allows to reduce uncertainties (Figure 3). The constrained $C_{ant}$ uptake of $31.5 \pm 0.9$
Pg C is 0.9 Pg C smaller than the interior ocean $C_{ant}$ estimate of $34 \pm 4$ Pg C based on observations (Gruber et al., 2019a) when
subtracting the multi-model mean climate-driven $CO_2$ flux estimate from the CMIP6 models of 1.6 Pg C (Table A3). This
difference of 0.9 Pg C is smaller than the uncertainties. Furthermore, the constrained $C_{ant}$ uptake of $31.5 \pm 0.9$ Pg C is 2.5 Pg
C larger than the observation-based air-sea $C_{ant}$ flux estimates from 1994 to 2007 of $29 \pm 4$ Pg C from the Global Carbon
Budget 2021 (Table 1) but both estimates agree within the uncertainties. When comparing short period, for example the years
after 2013, the observation-based air-sea $C_{ant}$ flux estimates can deviate from the constrained CMIP6 ESM estimates (Figure

1) due to unforced climate variability-driven $CO_2$ flux. Thus, the small difference between observation-based ocean $C_{ant}$ uptake

estimates from 1994 to 2007 and the here provided results may not exist over a longer period of time and be caused by a

different timing and magnitude of decadal variabilities in ESMs and the real world (Landschützer et al., 2016; Gruber et al.,

2019b; Bennington et al., 2022), as well as uncertainties in the observation-based products (Bushinsky et al., 2019; Gloege et

al., 2021, 2022). Indeed, when the entire period for which observation-based air-sea $C_{ant}$ flux estimates from the Global Carbon

Budget are available (1990-2020), the constrained estimate of the ocean $C_{ant}$ sink based on ESMs (80.7 ± 2.5 Pg C) is very

similar as the observation-based estimate from surface ocean $pCO_2$ observations (81 ± 7 Pg C) (Table 1).

The good agreement between the air-sea $C_{ant}$ flux estimates from ESMs and surface ocean $pCO_2$ observations in combination

with interior ocean $C_{ant}$ of a similar magnitude suggests that the air-sea $C_{ant}$ flux from hindcast simulations over the last three

decades (68 ± 8 Pg C) and possibly also over the 1994-2007 period (26 ± 3 Pg C) underestimates the ocean $C_{ant}$ uptake (Table

1). Therefore, the Global Carbon Budget 2021 estimate of the ocean $C_{ant}$ uptake over the last decades, which is an average of

the estimate of $C_{ant}$ uptake from observation-based methods and hindcast models, should be corrected upwards. Reasons for

this underestimation may be an underestimation of the AMOC or the Southern Ocean inter-frontal sea surface salinity, an

overestimation of the Revelle factor, a too small ensemble of models (8 models) that is biased towards low uptake models, too

short spin-up times (Séférian et al., 2016), neglecting the water vapour pressure when calculating the local $pCO_2$ in each ocean

grid cell (Hauck et al., 2020) as is done in CMIP models (Orr et al., 2017), or different pre-industrial atmospheric $CO_2$ mixing

ratios (Bronselaer et al., 2017; Friedlingstein et al., 2022). However, even after correcting these hindcast simulations upwards

by employing the here identified emergent constraint, their corrected estimate may remain below the CMIP-derived estimate

for the period from 1994 to 2017 due to the historical decadal variations in the $C_{ant}$ uptake that is not represented with the same

phasing in fully coupled ESMs (Landschützer et al., 2016; Gruber et al., 2019b; Bennington et al., 2022). A detailed analysis

by the individual modelling teams would be necessary to identify the reason for underestimation in the individual hindcast

models as the output is not openly available.

Over the historical period from 1850 to 2020, the here identified constraint increases the simulated ocean $C_{ant}$ uptake by 15 Pg C ($r^2 = 0.80$) from $174 \pm 13$ Pg C to $189 \pm 7$ Pg C (Table 1). The constrained estimate of the $C_{ant}$ agrees within the uncertainties with the estimate from the Global Carbon Budget for the same period ($170\pm35$ Pg C) (Friedlingstein et al., 2022), which is a combination of prognostic approaches until 1959 (Khatiwala et al., 2013; DeVries, 2014), and ocean hindcast simulations and observation-based $CO_2$ flux products from 1960 to 2020 (Friedlingstein et al., 2022). However, our new estimate is 19 Pg C larger and could explain around three quarters of the budget imbalance ($B_{IM}$) between global $CO_2$ emissions and sinks over the period 1850 to 2020 (25 Pg C) (Friedlingstein et al., 2022) and contribute to answering an important outstanding question in the carbon cycle community.

Overall, this new estimate of the ocean $C_{ant}$ uptake, based on ESMs and constrained by observations, presents an independent and new estimate of the past and present ocean $C_{ant}$ uptake that is around 10% larger and 42-59% less uncertain than the multi-model average and its standard deviation, respectively. The lower bound of the uncertainty correction is for the past ocean $C_{ant}$ uptake since 1765 where the late-starting date correction introduces an uncertainty that cannot be reduced without running the simulations from 1765 onwards. Towards the end of the 20[th] century, the uncertainty from this correction becomes smaller so that the emergent constraint can reduce uncertainties by almost 60%.

### 3.2.1 Southern Ocean

While the constraints were applied globally, they can also be applicable regionally as shown for the inter-frontal sea surface salinity in the Southern Ocean (Terhaar et al., 2021b). Here, we update the regional constraint in the Southern Ocean with the now additionally available ESMs and extent the constraint by adding the basin-wide averaged Revelle factor in the Southern Ocean as a second variable. For the period from 1765 to 2005, the simulated multi-model mean air-sea $C_{ant}$ flux that is adjusted for the late starting date is $63.5 \pm 6.1$ Pg C. Please note that the numbers here are for fluxes from 1765 to 2005 and are not the same as in Terhaar et al. (2021b), where fluxes from 1850 to 2005 were reported. The two-dimensional constraint shows a higher correlation coefficient ($r^2=0.70$) than the one-dimensional constraint when only the inter-frontal sea surface salinity is used as a predictor ($r^2=0.62$). Slight differences to Terhaar et al. (2021b) exist due to the additional ESMs that are by now

available. When exploiting this relationship with observations of the Southern Ocean Revelle factor (12.19±0.01) and the sea surface salinity, the best estimate of the cumulative air-sea $C_{ant}$ flux from 1765 to 2005 in the Southern Ocean increases to 72.0±3.4 Pg C. In comparison, observation-based estimates for the same period report 69.6±12.4 Pg C (Mikaloff Fletcher et al., 2006) and 72.1±12.6 Pg C (Gerber et al., 2009). The constrained thus reduces the uncertainty not only globally but also in the Southern Ocean by 44%.

### 3.2.2 Atlantic Ocean

As for the Southern Ocean, we also apply a two-dimensional constraint to the Atlantic Ocean, using the AMOC and the basin-wide averaged surface ocean Revelle factor in the North Atlantic as predictor. The unconstrained cumulative air-sea $C_{ant}$ flux from 1765 to 2005 in the North Atlantic adjusted for the late starting date is 21.9 ± 3.3 Pg C. For this period, the two-dimensional constraint results in a relationship with a correlation coefficient of 0.57. If only the AMOC had been used the correlation factor would have been 0.49. When exploiting this relationship with observations of the North Atlantic Revelle factor and AMOC, the best estimate of the cumulative air-sea $C_{ant}$ flux from 1765 to 2005 in the Atlantic Ocean increases to 22.7±2.2 Pg C. In comparison, observation-based estimates are 20.4±4.9 Pg C (Mikaloff Fletcher et al., 2006) and 20.4±6.5 Pg C (Gerber et al., 2009). The constrained and unconstrained estimates are both above the observation-based estimates but within the uncertainties. The constrained estimate is even higher than the unconstrained one, but only by 0.8 Pg C, and its uncertainty is reduced by 33%.

### 4 Consequences for projected ocean anthropogenic carbon uptake and acidification over the 21[st] century

As the present and future $C_{ant}$ uptake are strongly correlated across ESMs, the here identified relationship can also be used to constrain future projections of the global ocean $C_{ant}$ uptake. The global ocean $C_{ant}$ uptake from 2020 to 2100 increases from 156 ± 11 Pg C to 173 ± 8 Pg C ($r^2$=0.56) under the high-mitigation low emissions Shared Socioeconomic Pathway 1-2.6 (SSP1-2.6) that likely allows to keep global warming below 2°C (O'Neill et al., 2016; Riahi et al., 2017), from 251 ± 17 Pg C to 277 ± 9 Pg C ($r^2$=0.74) under the middle-of-the-road SSP2-4.5, and from 407 ± 30 Pg C to 445 ± 12 Pg C ($r^2$=0.87) under the high-emissions no mitigation SSP5-8.5 (Figure 1b). Overall, the future ocean $C_{ant}$ uptake in CMIP6 models is thus 9-11% larger

than simulated by ESMs and 32-62% less uncertain depending on the future scenario. The correlation coefficient and hence the uncertainty reduction reduces, but remains still large, when atmospheric $CO_2$ stops to increase (SSP1-2.6, SSP2-4.5). Larger uncertainties for stabilization than for near-exponential growth scenarios are expected as the reversal of the atmospheric $CO_2$ growth rate will exert a stronger external impact on the magnitude of the ocean carbon sink (McKinley et al., 2020).

The increase in projected uptake of $C_{ant}$ also increases the estimate of future ocean acidification rate. For ocean ecosystems, the threshold for water masses become undersaturated towards specific calcium carbonate minerals ($\Omega=1$) is of critical importance (Orr et al., 2005; Fabry et al., 2008; Doney et al., 2020), although negative effects for some calcifying organisms can already be observed at saturation states above one (Ries et al., 2009) and some calcifying organisms can even live in undersaturated waters (Lebrato et al., 2016). Over the 21$^{st}$ century, the volume of water masses in the global ocean that remain supersaturated towards the meta-stable calcium carbonate mineral aragonite is projected to decrease in CMIP6 from 283 million km$^3$ in 2002 (based on GLODAPv2 observations (Lauvset et al., 2016)) to 194±6 million km$^3$ under SSP1-2.6, to 143±4 million km$^3$ under SSP2-4.5, and to 97±4 million km$^3$ under SSP5-8.5. The constraint reduces these estimates to 186±5, 138±2, and 93±2 million km$^3$ respectively (r$^2$=0.31-0.69), resulting in an additional decrease of the available habitat for calcifying organisms of 3.7-7.4 million km$^3$ depending on the scenario. This additionally projected habitat loss is mainly located in the mesopelagic layer between 200 m and 1000 m and affects thus organisms that live their permanently or temporarily during diel vertical migration (Behrenfeld et al., 2019). The additionally undersaturated volume corresponds to an area of 1.6-3.1 times the area of the Mediterranean Sea whose mesopelagic layer would be additionally undersaturated towards aragonite. However, the global character of the constraint and the uncertainty of the interior distribution of $C_{ant}$ do not allow to localise these areas.

## 5 Robustness of the emergent constraint and possible impact of changing riverine carbon input over time

Emergent constraints across large datasets such as an ensemble of ESMs with hundreds of variables can always be found and might not necessarily be reliable and robust (Caldwell et al., 2014; Brient, 2020; Sanderson et al., 2021; Williamson et al., 2021). To test the robustness of emergent constraints, three criteria were proposed (Hall et al., 2019). The constraint must be

relying on well understood mechanisms, that mechanism must be reliable, and the constraint must be validated in an independent model ensemble.

Here, the well understood mechanisms are the fundamental ocean biogeochemical properties such as the Revelle factor (Revelle and Suess, 1957), as well as the Southern Ocean and North Atlantic large-scale ocean circulation features that are known to be the determining factors for the ocean ventilation (Marshall and Speer, 2012; Talley, 2013; Buckley and Marshall, 2016). For the Southern Ocean, the verification of the link between sea surface salinity and $C_{ant}$ uptake was previously done by linking the sea surface salinity, to the density, and to the volume of intermediate and mode waters in each model. Furthermore, the robustness of the constraint was tested against changes in the definition of the inter-frontal zone (Terhaar et al., 2021b). In addition, other potential predictors were tested, such as the magnitude and seasonal cycle of sea-ice extent, wind curl, and the mixed layer depth, and upwelling strength of circumpolar deep waters. All these variables are known to influence air-sea gas exchange, freshwater fluxes, and circulation and, in turn, salinity and $C_{ant}$ uptake. However, none of these factors alone explains biases in the surface salinity and $C_{ant}$ uptake in the Southern Ocean. Therefore, the sea surface salinity that emerges as a result of all these individual processes represents, so far, the best variable in terms of mechanistic explanation and observational uncertainty to bias-correct models for Southern Ocean $C_{ant}$ uptake. Further evidence for the underlying mechanism of the relationship between Southern Ocean sea surface salinity and $C_{ant}$ uptake was provided by a later study that analysed explicitly the stratification in the water column (Bourgeois et al., 2022). Here, we further showed that the Southern Ocean $C_{ant}$ uptake constrained by the Revelle factor and the inter-frontal sea surface salinity compares much better to observation-based estimates than the unconstrained estimate, further corroborating the identified regional constraint and mechanism (section 3.2.1).

Similarly, it was shown that the transport of $C_{ant}$ by the AMOC is crucial for the $C_{ant}$ uptake in the North Atlantic (Winton et al., 2013; Goris et al., 2018; Brown et al., 2021). As the AMOC is predominantly observed at 26.5°N, a change to the definition is not possible. Instead, we replaced the AMOC as a predictor by another indicator for deep-water formation, namely the area of waters in the North Atlantic below which the water column is weakly stratified (see Appendix A.1 and Table A4) (Hess,

2022). The results remain almost unchanged, indicating the robustness of the constraint and that the AMOC is indeed a good indicator for the stability of the water column in the North Atlantic and the associated deep-water formation. As for the Southern Ocean, we also made a regional two-dimensional constraint using the AMOC and the regional Revelle factor and compared it to observation-based $C_{ant}$ flux estimates. The good relationship between the AMOC and the North Atlanic $C_{ant}$

uptake improves the confidence in the AMOC as a valid predictor.

Eventually, we have also tested the robustness of the biogeochemical predictor, by varying the definition of the Revelle factor. First, the Revelle factor was only calculated north of 45°N and south of 45°S, assuming that the high-latitude regions are responsible for the largest $C_{ant}$ uptake, and second, the global Revelle factor was calculated by weighting the Revelle factor in

each cell by the multi-model mean cumulative $C_{ant}$ uptake from 1850 to 2100 in that cell so that the Revelle factor in cells with larger uptake is more strongly weighted. Under both definitions, the results remain almost unchanged (Table A4). Furthermore, the Revelle factor has been shown here to improve the $C_{ant}$ uptake in the Atlantic and Southern Ocean and has been earlier shown to determine the $C_{ant}$ uptake in the tropical Pacific Ocean (Vaittinada Ayar et al., 2022), suggesting that the Revelle factor is a robust predictor of global and regional ocean $C_{ant}$ uptake.

To provide further indication for the importance of the AMOC and the Southern Ocean surface salinity and the three-dimensional constraint in general, we have compared simulated CFC-11, provided by 10 ESMs from CMIP6, with observed CFC-11 from GLODAPv2.2021 (Lauvset et al., 2021) (Appendix A.4) and also compared the interior ocean distribution of $C_{ant}$ with observation-based estimates (Sabine et al., 2004; Gruber et al., 2019a) (Appendix A.5). The comparison of CFCs

demonstrates the importance of the AMOC for the ventilation of the North Atlantic, as ESMs with a low AMOC underestimate the observed subsurface CFC-11 concentrations in the North Atlantic. Similarly, ESMs with a small inter-frontal Southern Ocean surface salinity underestimate observed subsurface (below 200 m) CFC-11 concentrations in the Southern hemisphere. In addition to the evaluation with observations of CFC, the comparison of the interior ocean $C_{ant}$ distribution demonstrates first that the ESMs on average represent the observation-based distributions within the margins of error (Tables A5 and A6). Only

in the Southern hemisphere, the ESM average remains below, as expected due to the average ESM bias towards too low inter-

frontal sea surface salinities, too little formation of mode and intermediate waters, and hence too little storage of $C_{ant}$ in the Southern hemisphere. When using the model that represents best the three predictors, GFDL-ESM4 (Dunne et al., 2020; Stock et al., 2020), the comparison to observation-based interior ocean $C_{ant}$ distribution becomes almost identical (Tables A7 and A8), suggesting that a better representation of these parameters indeed improves the simulation of $C_{ant}$ uptake and its distribution in the ocean interior.

To validate the here identified constraint in another model ensemble, we used all six ESMs of the CMIP5 ensemble that provided all necessary output variables (Table A1). As these six ESMs are not sufficient to robustly fit a function with four unknown parameters, we applied the predicted relationship by the CMIP6 models to the CMIP5 models and evaluated how well this relationship allows to predict the simulated historical $C_{ant}$ uptake by these models. The CMIP6 derived relationship allows to predict the simulated $C_{ant}$ uptake with an accuracy of 3% (±5 Pg C) for the period from 1850 to 2014 and with an accuracy of 4% (±1.3 Pg C) for the period from 1994 to 2007 (Figure A5). The largest uncertainty stems from the NorESM2-ME model, which simulates a historical AMOC strength of ~30 Sv, almost twice as large as the observed AMOC strength and ~9 Sv larger than all other CMIP6 ESMs over which the relationship was fitted. For such strong deviations from the observations and other ESMs, the linear relationship might not be applicable anymore. However, despite one out of six ESMs from CMIP5 having a particularly high AMOC, the here identified relationship still allows to predict the simulated $C_{ant}$ uptake with small uncertainties and hence confirms its applicability.

Despite this robustness, emergent constraints are, by definition, always relying on the existing ESMs and on the processes that are represented by these ESMs. If certain processes are not implemented or implemented in the same way across all ESMs, biases over the entire model ensemble can occur that cannot be corrected by an emergent constraint (Sanderson et al., 2021). Possible non-represented processes in our case are among others changing freshwater input from the Greenland and Antarctic ice sheet that may impact the freshwater cycle and circulation in the Southern Ocean or the AMOC, and changes in riverine input of carbon over time. However, the expected effect of ice melt on sea surface salinity in the Southern Ocean and on the AMOC is small compared to the model spread (Bakker et al., 2016; Terhaar et al., 2021b), at least on the timescales considered

here. Changing riverine carbon fluxes could, however, have a larger effect. So far, only one CMIP6 ESM, the CNRM-ESM2-1 (Séférian et al., 2019), has dynamic carbon riverine delivery that changes with global warming. In this model, carbon riverine delivery increases over the $20^{st}$ century so that the interior ocean change in $C_{ant}$ in 2000 is around 19 Pg C smaller than the air-sea $C_{ant}$ uptake (Figure A4). The situation reverses at the beginning of the $21^{st}$ century, so that riverine carbon delivery increases

and the interior ocean change in $C_{ant}$ becomes up to 60 Pg C larger than the air-sea $C_{ant}$ uptake. As such, riverine carbon delivery has the potential to enhance or decrease the ocean $C_{ant}$ inventory in addition to air-sea $C_{ant}$ uptake. This would also question the comparability of $C_{ant}$ inventory and air-sea $C_{ant}$ uptake estimates. However, the present state of the ESMs does not allow a quantitative assessment of this process and future research is needed.

In addition, parametrizations of non-represented processes such as mesoscale and sub-mesoscale circulation features like small-scale eddies may lead to biases in the model ensemble. For individual models, it has been shown that changes in horizontal resolution and hence a more explicitly simulated circulation change the model physics and biogeochemistry, and hence also the ocean carbon and heat uptake (Lachkar et al., 2007, 2009; Dufour et al., 2015; Griffies et al., 2015). However, an increase in resolution does not necessarily lead to improved simulations and the changes in oceanic $C_{ant}$ uptake maybe lower

or higher, depending on the model applied. When increasing the NEMO ocean model from a non-eddying version (2° horizontal resolution) to an eddying version (0.5°), Lachkar et al. (2009) find a decrease in the sea surface salinity by around 0.1 at the Southern Ocean surface that brings the model further away from the observed salinity, a decrease of the volume of Antarctic intermediate water and a decrease in the Southern Ocean uptake of CFC and hence likely also of $C_{ant}$. This example corroborates the underlying mechanism of the emergent constraint in the Southern Ocean that higher sea surface salinity

directly affects the formation of Antarctic intermediate water and the uptake of $C_{ant}$. Another example can be found within the ESM ensemble of CMIP6. The MPI-ESM-1-2-HR and MPI-ESM-1-2-LR have a horizontal resolution of 0.4° and 1.5° respectively but the same underlying ocean model. The high-resolution version has an inter-frontal salinity of 33.98, a Southern Ocean surface Revelle factor of 12.82, and a Southern Ocean $C_{ant}$ uptake from 1850 to 2005 of 56.4 Pg C. The coarser resolution version has an inter-frontal sea surface salinity of 33.92, a Southern Ocean surface Revelle factor of 12.89, and a Southern

Ocean $C_{ant}$ uptake of 58.0 Pg C. These differences are much smaller than the inter-model differences (33.66-34.15 for salinity,

12.14-13.11 for the Revelle factor, and 48.8-71.1 Pg C for the Southern Ocean $C_{ant}$ uptake) that result from different ocean circulation and biogeochemical models, sea ice models, and atmospheric and land biosphere models, as well as the coupling between these models. These examples show that higher resolution does not necessarily lead to better results, effects potentially the predictor and the predicted variable in the same way, and that differences in the underlying model components and spin-up and initialization strategies lead so far to much larger differences between ESMs than resolution does(Séférian et al., 2020). As long as simulations with higher resolution, which are also spun-up over hundreds of years (Séférian et al., 2016), are not yet available, and potentially important processes such as changing riverine fluxes and freshwater from land ice are not included, it remains speculative if higher resolution would lead to a reduction of inter-model uncertainty, or even a better representation of the observations. Moreover, the here-identified relationships that are based on the current understanding of physical and biogeochemical oceanography and that were tested for robustness in several ways may likely also exist across ensembles of eddy-resolving models.

**6 Conclusion**

The here identified three-dimensional emergent constraint allows identifying a bias towards too low $C_{ant}$ uptake by ESMs from CMIP6, reduced uncertainties of the global ocean $C_{ant}$ sink, and led to an enhanced process understanding of the $C_{ant}$ uptake in ESMs. The constraint was tested for robustness in multiple ways and across different model ensembles. It was evaluated regionally and globally against CFC measurements, estimates of the interior ocean $C_{ant}$ accumulation, and against observation-based estimates of the air-sea $CO_2$ flux globally and regionally. The constraint demonstrates that the global ocean $C_{ant}$ uptake can be estimated from three observable variables, the salinity in the subtropical-polar frontal zone in the Southern Ocean, the Atlantic Meridional Overturning Circulation, and the global surface ocean Revelle factor. The uncertainties of the regional ocean $C_{ant}$ uptake estimates in the Atlantic and Southern Ocean can also be reduced with the respective regional predictors. Improved or continuing observations of these quantities (Lauvset et al., 2016; Zweng et al., 2018; Locarnini et al., 2018; Claustre et al., 2020; McCarthy et al., 2020) and their representation and evaluation in ESMs and ocean models should therefore be of priority in the next years and decades. Although biogeochemical variables were tuned or calibrated in more ESMs in

CMIP6 than in CMIP5 (Séférian et al., 2020), this tuning does not seem to result in better results than in untuned ESMs yet (Figure A3).

Moreover, biases in these quantities and corrections for the late starting date may well be the reason for offset between models and observations over the last 30 years (Hauck et al., 2020; Friedlingstein et al., 2022). Although the here identified constraints cannot correct for misrepresentation of the unforced decadal variability, such variability plays likely a minor role when averaging results over longer periods. Indeed, we find good agreement between our estimate and the observation-based estimate from the Global Carbon Budget 2021 for the period from 1990 to 2020. This agreement suggests that the hindcast models underestimate the ocean $C_{ant}$ uptake. This underestimation is thus likely the explanation for the difference between models and observation-based product in the Global Carbon Budget (Friedlingstein et al., 2022). However, the output of the Global Carbon Budget hindcast models is not publicly available for evaluating possible data-model differences for the inter-frontal sea surface salinity, the AMOC, and the Revelle factor.

Despite this step forward in the understanding of ESMs, a comprehensive research strategy that combines the measurements of important physical, biogeochemical, and biological parameters in the ocean with other data streams and modelling is needed. A comprehensive approach is necessary to improve our still incomplete understanding of the global carbon cycle and its functioning in the climate and Earth system over the past and under ongoing global warming.

The larger than previously estimated future ocean $C_{ant}$ sink corresponds to around 2 to 4 years of present-day $CO_2$ emissions (~10.5 Pg C yr$^{-1}$) depending on the emissions pathway. The larger ocean $C_{ant}$ sink thus increases the estimated remaining emission budget, but only by a small amount. However, it also results in enhanced projected ocean acidification that may be harmful for large, unique ocean ecosystems (Fabry et al., 2008; Gruber et al., 2012; Kawaguchi et al., 2013; Kroeker et al., 2013; Doney et al., 2020; Hauri et al., 2021; Terhaar et al., 2021a).

This study follows recent approaches by the IPCC and climate science that suggest using the best available information about models instead of a multi-model mean to provide consistent and accurate information for climate science and policy (IPCC, 2021; Hausfather et al., 2022). The here provided improved estimate of the size of the global ocean carbon sink may help to close the carbon budget imbalance since 1850 (Friedlingstein et al., 2022) and to improve the understanding of the overall carbon cycle and the global climate (IPCC, 2021). Eventually, a better understanding of the ocean carbon sink and the reduction of its uncertainties in the past and in the future allows better targeted climate and ocean policies (IPCC, 2022).

**Appendix A**

**A.1 Earth System Models**

Model output from 18 Earth System Models from CMIP6 and 6 Earth System Models from CMIP5 (Table A1) were used for the analyses.

**Table A1. CMIP5 and CMIP6 models used in this study and the corresponding model groups**

| Model name* | Modeling center | References |
|---|---|---|
| ACCESS-ESM1-5 | Commonwealth Scientific and Industrial Research Organisation (CSIRO) | (Ziehn et al., 2020) |
| *CanESM2* | | |
| CanESM5 | Canadian Centre for Climate Modelling and Analysis | (Chylek et al., 2011; Christian et al., 2022) |
| CanESM5-CanOE | | |
| *CESM1-BGC* | | (Gent et al., 2011; Lindsay et al., 2014; Danabasoglu et al., 2020) |
| CESM2 | Community Earth System Model Contributors | |
| CESM2-WACCM | | |
| CMCC-ESM2 | Centro Euro-Mediterraneo per I Cambiamenti Climatici | (Lovato et al., 2022) |
| CNRM-ESM2-1 | Centre National de Recherches Meteorologiques / Centre Europeen de Recherche et Formation Avancees en Calcul Scientifique | (Séférian et al., 2019) |
| EC-Earth3-CC | EC-Earth consortium (http://www.ec-earth.org/community/consortium/) | (Döscher et al., 2022) |
| *GFDL-ESM2M* | | (Dunne et al., 2012; Held et al., 2019; Dunne et al., 2020; Stock et al., 2020) |
| GFDL-CM4 | NOAA Geophysical Fluid Dynamics Laboratory (NOAA GFDL) | |
| GFDL-ESM4 | | |
| IPSL-CM6A-LR | Institut Pierre-Simon Laplace (IPSL) | (Boucher et al., 2020) |
| MIROC-ES2L | Japan Agency for Marine-Earth Science and Technology, Atmosphere and Ocean Research Institute (The University of Tokyo), and National Institute for Environmental Studies | (Hajima et al., 2020) |
| *MPI-ESM-LR* | | (Giorgetta et al., 2013; Mauritsen et al., 2019; Gutjahr et al., 2019) |
| *MPI-ESM-MR* | Max-Planck-Institut für Meteorologie (Max Planck Institute for Meteorology) | |
| MPI-ESM-1-2-LR | | |
| MPI-ESM-1-2-HR | | |
| MRI-ESM2-0 | Meteorological Research Institute (Japan Meteorological Agency) | (Yukimoto et al., 2019) |
| *NorESM1-ME* | | |
| NorESM2-LM | Norwegian Climate Centre | (Bentsen et al., 2013; Tjiputra et al., 2020) |
| NorESM2-MM | | |
| UKESM1-0-LL | Met Office Hadley Centre | (Sellar et al., 2020) |

*CMIP5 models are written in italics

The analysed variables include the air-sea $CO_2$ flux (fgco2, name of the variable in standardized CMIP output), total dissolved inorganic carbon (dissic), total alkalinity (talk), total dissolved inorganic silicon (si), total dissolved inorganic phosphorus (po4), potential temperature (thetao), salinity (so), and the Atlantic meriodional streamfunction (msftmz or msftyz). All ESMs were included for which the entire set of variables was available on the website of the Earth System Grid Federation at the start of the analysis. Based on these variables, all other presented variables were derived:

- - The air-sea $C_{ant}$ flux was calculated as the difference in air-sea $CO_2$ flux between the historical plus future (SSP for CMIP6 and RCP for CMIP5) simulation and the correspondent pre-industrial control simulation on the native model grids (where possible). The air-sea $C_{ant}$ fluxes were corrected for their late starting date in 1850 (and 1861 for GFDL-ESM2M) and the slightly higher atmospheric $CO_2$ mixing ratio in that year compared to the beginning of the industrialization and the start of the $CO_2$ increase in 1765 (Bronselaer et al., 2017). To that end, we scaled the simulated air-sea $C_{ant}$ flux with the anthropogenic change in the atmospheric partial pressure of $CO_2$ ($pCO_2$) with respect to pre-industrial conditions following previous studies (Mikaloff Fletcher et al., 2006; Gruber et al., 2009; Terhaar et al., 2021b):

$$C_{ant}^{corr}(t) = C_{ant}(t) \frac{pCO_2(t) - pCO_2(1765)}{pCO_2(t) - pCO_2(1850)}, \tag{3}$$

   with $C_{ant}(t)$ being the simulated air-sea $C_{ant}$ flux by the respective ESM in year t and $C_{ant}^{corr}(t)$ being the corrected air-sea $C_{ant}$ flux. For GFDL-ESM2M, which starts in 1861, the correction was made with respect to $pCO_2(1861)$. When $pCO_2(t)$ is close to $pCO_2(1850)$, their difference becomes unrealistically large, causing overly strong flux corrections. Therefore, we limited the flux correction in magnitude using the correction term in year 1950 as an upper limit. By doing so, we do not only remove unrealistically high air-sea $C_{ant}$ fluxes before 1950 but also reach excellent agreement with the previously estimated air-sea $C_{ant}$ fluxes correction term by Bronselaer et al. (2017) (Figure A1). When the cumulative $C_{ant}$ fluxes since 1765 are shown, an additional amount of 12 Pg C (16 Pg C for GFDL-ESM2M) was added that was estimated to have entered the ocean before 1850 (Bronselaer et al., 2017). For comparison, we

also calculated the constrained estimates for the ocean $C_{ant}$ sink when no air-sea $C_{ant}$ flux correction is applied (Table A2). Bronselaer et al. (2017) estimate the uncertainty of the correction to be ±16% for cumulative $C_{ant}$ fluxes from 1765 to 1995. Although uncertainties reduce over time, we apply the 16% from the past to all estimates and hence

provide a conservative upper bound of this uncertainty.

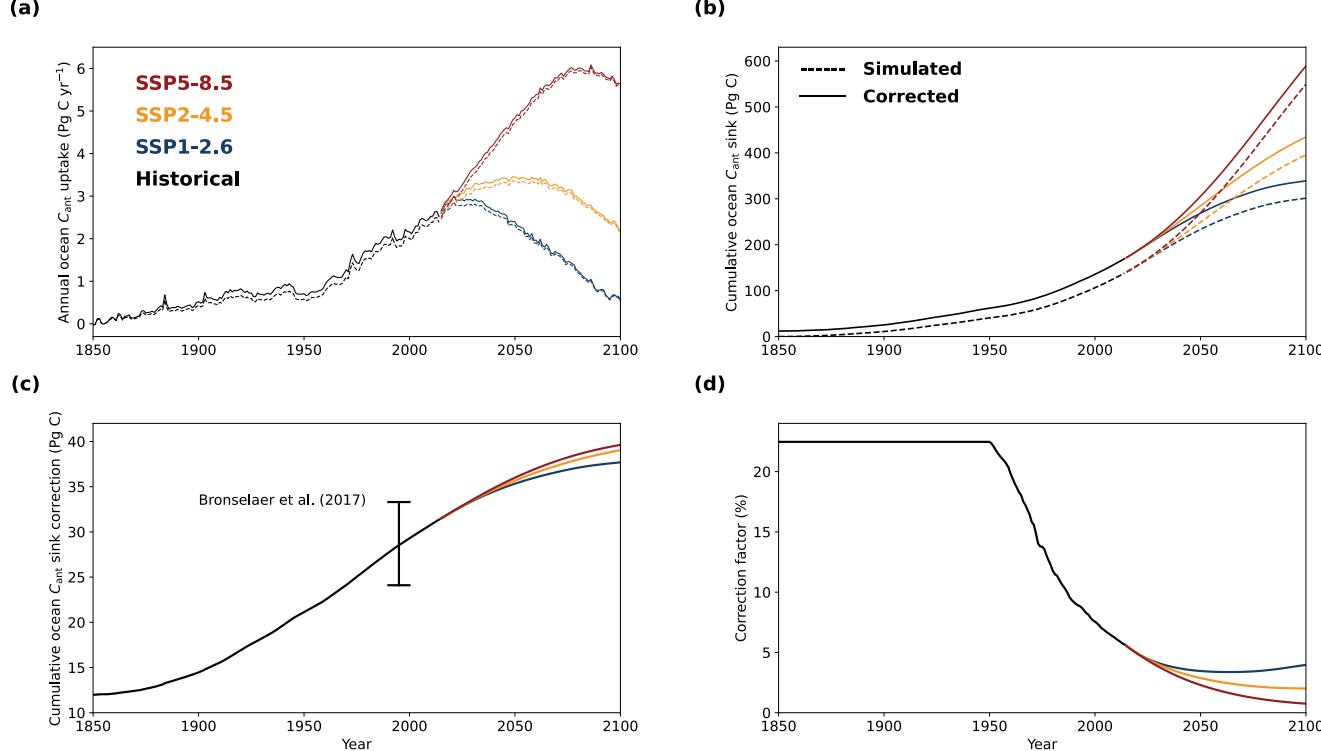

**Figure A1. Correction of simulated anthropogenic carbon air-sea flux for the late starting date in Earth System Models.** Multi-model **a)** annual mean anthropogenic carbon ($C_{ant}$) air-sea flux for 17 ESMs from CMIP6 before (dashed lines) and after (solid lines) the correction

for the late starting date over the historical period from 1850 to 2014 (black) and for the future from 2015 to 2100 under SSP1-2.6 (blue), SSP2-4.5 (orange), and SSP5-8.5 (red). **b)** Cumulative ocean $C_{ant}$ uptake since 1765 (corrected simulated flux) and 1850 (raw simulated flux), **c)** difference between cumulative ocean $C_{ant}$ uptake between corrected and raw simulated flux, and **d)** the correction factor that was applied. The $C_{ant}$ correction that was estimated by Bronselaer et al. (2017) is shown for in **c)**. The cumulative $C_{ant}$ uptake from 1765 to 1850 was set to 12 Pg C as estimated by Bronselaer et al. (2017).

**Table A2. Global ocean air-sea $CO_2$ flux estimates based on 17 ESMs from CMIP6 before and after constraint over different periods with corrected and uncorrected estimates and with and without CNRM-ESM2-1. Prior uncertainty is the multi-model standard deviation and constrained uncertainty is a combination of the multi-model standard deviation after correction and the uncertainty from the correction itself (see section 3.1).**

| Period | Cumulative air-sea $C_{ant}$ flux (Pg C) | | | | | |
|---|---|---|---|---|---|---|
| | Raw simulated | | Starting date corrected | | Corrected + CNRM-ESM2-1 | |
| | Prior | Constrained | Prior | Constrained | Prior | Constrained |
| 1994-2007 | $26.8 \pm 2.1$ | $29.3 \pm 0.8$ | $28.8 \pm 2.2$ | $31.5 \pm 0.9$ | $28.6 \pm 2.3$ | $31.3 \pm 1.2$ |
| 1850-2014 | $138 \pm 10$ | $150 \pm 5$ | $157 \pm 12$ | $171 \pm 5$ | $156 \pm 12$ | $171 \pm 6$ |
| 1850-2020 | $154 \pm 11$ | $167 \pm 5$ | $174 \pm 13$ | $189 \pm 6$ | $173 \pm 13$ | $189 \pm 6$ |
| 2020-2100 (SSP1-2.6) | $150 \pm 11$ | $167 \pm 7$ | $156 \pm 11$ | $173 \pm 7$ | $156 \pm 11$ | $173 \pm 7$ |
| 2020-2100 (SSP2-4.5) | $244 \pm 16$ | $269 \pm 8$ | $251 \pm 17$ | $277 \pm 9$ | $251 \pm 16$ | $276 \pm 9$ |
| 2020-2100 (SSP5-8.5) | $399 \pm 29$ | $436 \pm 11$ | $407 \pm 30$ | $445 \pm 11$ | $405 \pm 29$ | $444 \pm 12$ |

- Accordingly, the change in ocean interior $C_{ant}$ was calculated as the difference in total dissolved inorganic carbon between the historical plus future (SSP/RCP) simulation and the correspondent pre-industrial control simulation on the native model grids (where possible).

- The change in air-sea $CO_2$ flux that is caused by a changing climate was calculated as the difference in fgco2 in the historical simulation and the 'bgc' simulation in which only atmospheric $CO_2$ changes, but not the climate. These 'bgc' simulations were available for 5 ESMs (Table A3)

**Table A3. Climate-driven changes in the air-sea CO$_2$ flux (Pg C yr$^{-1}$) as simulated by 5 Earth System Models from CMIP6**

| Year | Climate-driven changes in the cumulative air-sea CO$_2$ flux (Pg C) | | | | | | |
|---|---|---|---|---|---|---|---|
| | ACCESS-ESM1-5 | CanESM5 | MIROC-ES2L | MRI-ESM2-0 | NorESM2-LM | Multi-model mean | Multi-model standard deviation |
| 1994-2007 | -1.7 | -1.7 | -1.4 | -2.2 | -0.7 | -1.6 | 0.5 |

.

- The surface ocean Revelle factor was calculated from sea surface total dissolved inorganic carbon (dissic), total alkalinity (talk), total dissolved inorganic silicon (si), total dissolved inorganic phosphorus (po4), potential temperature (thetao), and salinity (so) averaged around the year 2002 (from 1997 to 2007 for CMIP6 and 1999 to 2005 for CMIP5; 2005 is the last year of the historical simulation) using *mocsy2.0* (Orr and Epitalon, 2015) with its default constants that are recommended for best practice (Dickson et al., 2007). The years were centred around 2002 to make the Revelle factor comparable to the one estimated based on GLODAPv2, which is normalized to the year 2002 (Lauvset et al., 2016). As the Revelle factor describes the relative change in $C_T$ per relative change in $pCO_2$ (Revelle and Suess, 1957), the absolute uptake of $C_T$ does not only depend on the Revelle factor but also on the natural $C_T$ in the surface ocean. To calculate the buffer capacity for each ESM, the Revelle factor was therefore adjusted in each grid cell by multiplying it by the ratio of observed $C_T$ and the simulated $C_T$ in each ESM separately. Data from each ESM was regridded on a regular 1°x1° grid to make it comparable to the gridded GLODAPv2 data. Furthermore, a mask was applied before the basin-wide averaged Revelle factor was calculated so that only values were used where all ESMs and the gridded GLODAPv2 product had data. In addition, marginal seas (Mediterranean Sea, Hudson Bay, Baltic Sea) were excluded because global ESMs are not designed to accurately represent these small-scale seas. In addition, the surface ocean carbonate ion ($CO_3^{2-}$) concentration was calculated that the $C_T$-adjusted Revelle factor is mainly determined by the $CO_3^{2-}$ concentrations, which itself can be approximated by the difference between surface ocean alkalinity and $C_T$ (Figure A2).

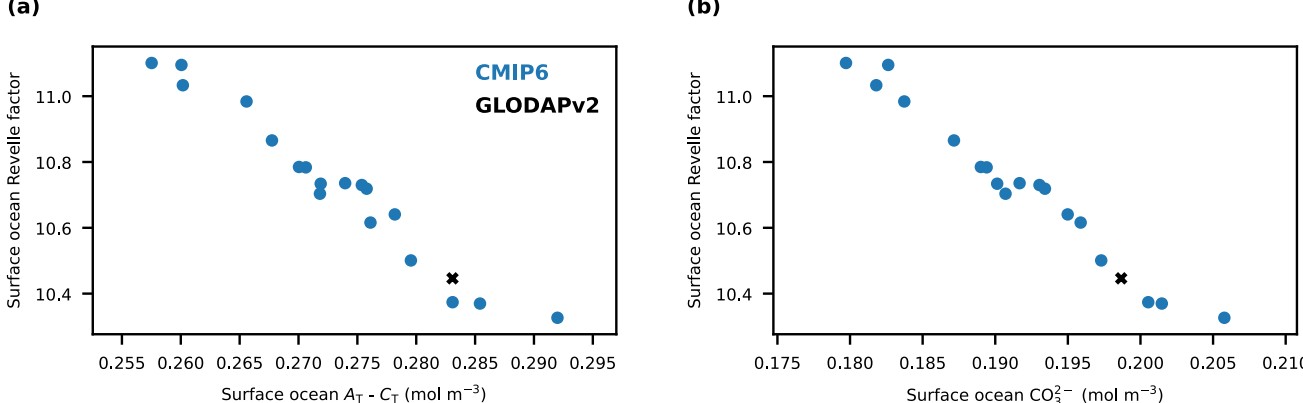

**Figure A2. Surface ocean Revelle factor against the difference of surface alkalinity and dissolved inorganic carbon, and against surface carbonate ion concentrations.** Basin-wide averaged surface ocean Revelle factor as simulated by 18 ESMs from CMIP6 (blue dots) against the basin-wide averaged surface ocean **a)** the difference between total alkalinity ($A_T$) and $C_T$, and **b)** carbonate ion ($CO_3^{2-}$) concentrations. The observation-based estimates from GLODAPv2 are shown as black crosses. The Revelle factor in each ESM was adjusted for biases in the surface ocean $C_T$ (see Appendix A.1).

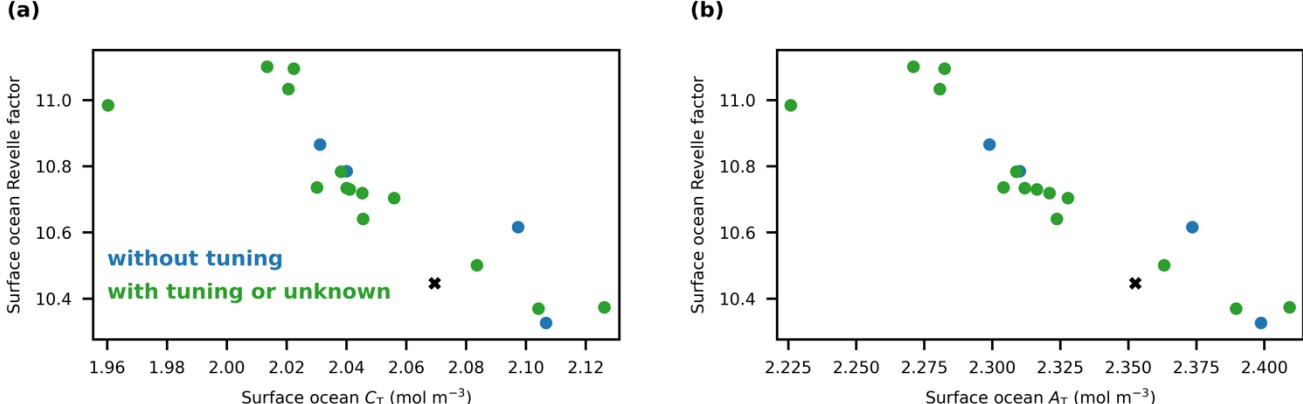

**Figure A3. Surface ocean Revelle factor against the surface alkalinity and dissolved inorganic carbon.** Basin-wide averaged surface ocean Revelle factor as simulated by 18 ESMs from CMIP6 (blue dots) against the basin-wide averaged surface ocean **a)** total alkalinity ($A_T$) and **b)** $C_T$. The observation-based estimates from GLODAPv2 are shown as black crosses. The Revelle factor in each ESM was adjusted for biases in the surface ocean $C_T$ (see Appendix A.1).

- The monthly AMOC strength was calculated as the maximum of the streamfunction below 500 m at the latitude in the respective model that is closest to 26.5°N for each month from 2004 to 2020. After 2014, simulated output from SSP5-8.5 and RCP4.5 were used as all ESMs provided output for these pathways. For SSP5-8.5, the mole fraction of atmospheric $CO_2$ in SSP5-8.5 is 414.9 ppm in 2020 (Meinshausen et al., 2020), 2.5 ppm over the observed mole fraction of atmospheric $CO_2$ in 2020 (Trends in Atmospheric Carbon Dioxide (NOAA/GML)). For RCP4.5, the mole fraction of atmospheric $CO_2$ is 412.4 ppm in 2020. Such small differences in the mole fraction of atmospheric $CO_2$ do not cause detectable changes in global warming or the AMOC (IPCC, 2021).

- Future saturation states of aragonite were calculated from simulated changes in total dissolved inorganic carbon (dissic), total alkalinity (talk), total dissolved inorganic silicon (si), total dissolved inorganic phosphorus (po4), potential temperature (thetao) and salinity (so) since 2002 that are added to the respective observed variables from the gridded GLODAPv2 product, which are normalized to 2002, using *mocsy2.0* (Orr and Epitalon, 2015) with its default constants that are recommended for best practice (Dickson et al., 2007). By only adding simulated difference, model uncertainties in the initial state of the ocean biogeochemical system in the deeper ocean are removed (Orr et al., 2005; Terhaar et al., 2020a, 2021a, b). All variables were regridded before on a regular 1°x1° grid so that they could be added to the gridded GLODAPv2 data. The same mask that was also used to compare the Revelle factor was applied to make all projections comparable.

- The annual average sea surface salinity between the polar and subtropical front in the Southern Ocean was derived from regridded (1°x1° regular grid) monthly sea surface salinity and temperatures (for defining the fronts) following (Terhaar et al., 2021b).

- The area of weakly stratified waters was calculated based on climatologies of the potential temperature and salinity from 1995 to 2014 (Hess, 2022). All data was regridded on a regular 1°x1° grid with 33 depth levels before analysis. An area was defined as weakly stratified if the density gradient between the surface and the cell at 1000 m depth was smaller than 0.5 kg m$^{-3}$ in a given month, assuming that such a small monthly mean gradient allows mixing of water into the lower limb of the AMOC at some time in that month. This predictor, as well as the different ways of calculating the Revelle factor predictor (see section "Robustness of the emergent constraint and possible impact of

changing riverine carbon input over time"), was used to test the robustness of the here identified emergent constraint

(Table A4).

The model CNRM-ESM2-1 was not used for the constraints because it includes dynamical riverine forcing that no other model includes (Figure A4) and is not directly comparable. Instead, output from this ESM was prominently used in the section "Robustness of the emergent constraint and possible impact of changing riverine carbon input over time". However, even if CNRM-ESM2-1 had been included, the results change by less than 1% (Table A2).

**Table A4. Constrained global ocean air-sea CO$_2$ flux estimates based on 17 ESMs from CMIP6 with varying predictors.**

| Period | Cumulative air-sea $C_{ant}$ flux (Pg C) | | | |
|---|---|---|---|---|
| | Standard | Revelle factor | | Area of weakly stratified water column |
| | | >45°N & <45°S | Flux-weighted | |
| 1994-2007 | 31.5 ± 0.9 (r²=0.87) | 31.6 ± 1.1 (r²=0.80) | 31.7 ± 1.0 (r²=0.83) | 31.3 ± 1.1 (r²=0.78) |
| 1850-2014 | 171 ± 6 (r²=0.80) | 172 ± 8 (r²=0.65) | 173 ± 7 (r²=0.73) | 171 ± 7 (r²=0.74) |
| 1850-2020 | 189 ± 7 (r²=0.80) | 190 ± 8 (r²=0.64) | 191 ± 8 (r²=0.72) | 189 ± 7 (r²=0.73) |
| 2020-2100 (SSP1-2.6) | 173 ± 8 (r²=0.56) | 173 ± 8 (r²=0.56) | 172 ± 8 (r²=0.55) | 171 ± 8 (r²=0.53) |
| 2020-2100 (SSP2-4.5) | 277 ± 9 (r²=0.74) | 278 ± 9 (r²=0.71) | 277 ± 9 (r²=0.71) | 274 ± 9 (r²=0.72) |
| 2020-2100 (SSP5-8.5) | 445 ± 12 (r²=0.87) | 450 ± 13 (r²=0.83) | 449 ± 12 (r²=0.84) | 442 ± 12 (r²=0.84) |

**Figure A4. Anthropogenic carbon air-sea fluxes and inventory changes simulated by CNRM-ESM2-1. (a)** Cumulative air-sea anthropogenic carbon ($C_{ant}$) fluxes (solid lines) and $C_{ant}$ interior changes (dashed lines) as simulated by CNRM-ESM2-1 for the historic period until 2014 (black) and from 2015 to 2100 under SSP1-2.6 (blue), SSP2-4.5 (orange), and SSP5-8.5 (red), **(b)** as well as the difference of both quantities. The thin dashed black line in **(b)** indicates zero difference.

## A.2 Observations and observation-based products

Throughout this manuscript, three observation-based products are used to constrain the ESM output:

- Monthly climatologies of sea surface salinity and sea surface temperatures from the World Ocean Atlas 2018 (Zweng et al., 2018; Locarnini et al., 2018) were used to derive annual averages and uncertainties of the sea surface salinity between the polar and subtropical fronts in the Southern Ocean following Terhaar et al. (2021b). Climatologies of the World Ocean Atlas 2018 were also used to calculate the area of weakly stratified surface waters.

- Time series of the AMOC strength from the RAPID array (McCarthy et al., 2020) were used to calculate monthly means and uncertainties of the AMOC from 2004 to 2020.

- The gridded observation-based estimates of total dissolved inorganic carbon, total alkalinity, total dissolved inorganic silicon, total dissolved inorganic phosphorus, in-situ temperature, and salinity from GLODAPv2 (Lauvset et al., 2016) were used to calculate the Revelle factor and as a starting point for projected saturation states over the 21$^{st}$ century (see above).

## A.3 Validation of the identified constraint in CMIP5

The here identified emergent constraint was derived from an ensemble of 17 ESMs from CMIP6. To test the robustness of emergent constraints, these constraints should be validated in an independent ensemble of ESMs (Hall et al., 2019). Here, we used all 6 ESMs from CMIP5 that provided all necessary output variables for this analysis (see Appendix A.1). For all these models, the $C_{ant}$ uptake for the period from 1994 to 2007 and from 1850 to 2014 was predicted based on the simulated inter-frontal sea surface salinity in the Southern Ocean, the AMOC strength, and the global ocean basin-wide averaged Revelle factor using the multi-linear relationship derived from the CMIP6 models (Figure A5).

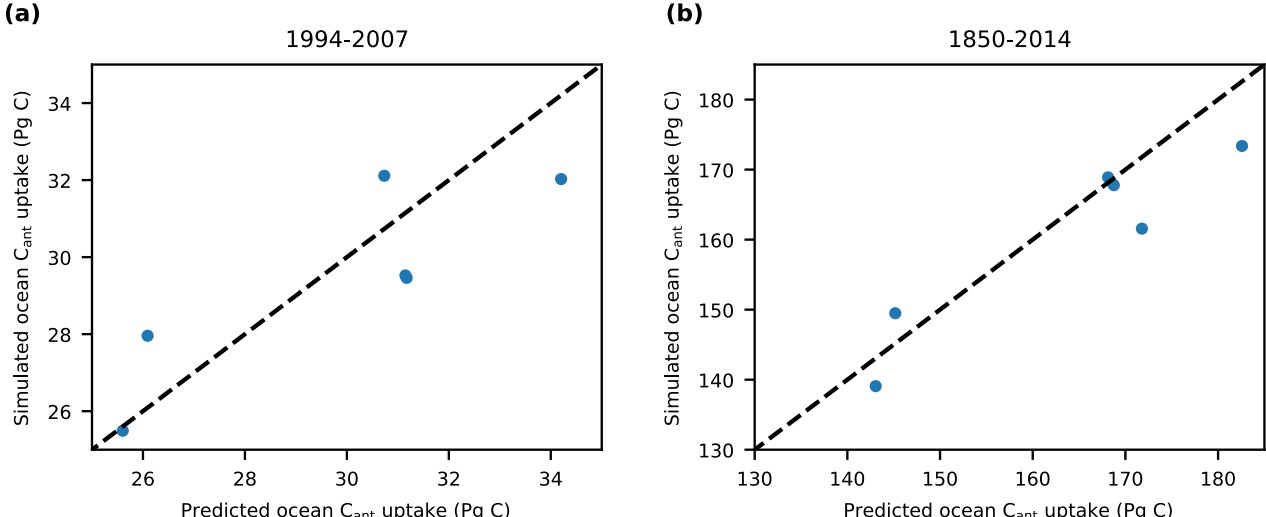

**Figure A5. Global ocean anthropogenic carbon uptake simulated by Earth System Models from CMIP5 against the predicted uptake based on simulated predictors from CMIP6 models.** Global ocean anthropogenic carbon uptake simulated by 6 ESMs from CMIP5 (Table A1) **a)** from 1994 to 2007 and **b)** from 1850 to 2014 against the predicted anthropogenic carbon uptake based on the simulated CMIP6 predictors in each ESM: the inter-frontal annual mean sea surface salinity in the Southern Ocean, the Atlantic Meridional Overturning Circulation, and the Revelle factor adjusted for surface ocean $C_T$. Please note that two ESMs are at almost the same place in **a)** with a predicted $C_{ant}$ uptake of around 31 Pg C.

## A.4 Comparison between simulated and observed CFC-11 concentrations

Comparison between simulated and observed CFC-11 uptake allows to estimate the ventilation of waters from the surface waters to the deeper ocean (Hall et al., 2002). Although CFCs can roughly evaluate the ventilation rate of the ocean, no perfect agreement between CFCs and $C_{ant}$ can be expected as CFCs are not taken up at the same speed as $C_{ant}$ (i.e., fast air-sea equilibration time scale for CFC) and their solubility has a different temperature dependency than the solubility of $C_{ant}$ (warm waters can hold less CFCs but more $C_{ant}$ due to their low Revelle factor, whereas cold waters hold more CFCs but less $C_{ant}$) (Revelle and Suess, 1957; Broecker and Peng, 1974; Weiss, 1974). These differences can lead to differences between uptake, storage, and distribution of CFCs and $C_{ant}$ that can become especially large in high-latitude oceans (Matear et al., 2003; Terhaar et al., 2020b).

Here, we use simulated CFC-11 from ESMs and observed CFC-11 from GLODAPv2.2021 (Lauvset et al., 2021) to provide

further evidence that the inter-frontal sea surface salinity in the Southern Ocean and the AMOC are good indicators for the

ocean ventilation and that ESMs tend to underestimate the ventilation of surface waters to the deeper ocean. Out of the 18

ESMs from CMIP6, 10 provided simulated 3D-fields of CFC-11 (CanESM5, CESM2, CESM2-WACCM, EC-Earth-CC,

GFDL-CM4, GFDL-ESM4, MRI-ESM2-0, NorESM2-LM, NorESM2-MM, UKESM1-0-LL). To compare these ESMs to the

observed concentrations, all ESMs were sampled at the same time (month and year), the same latitude and longitude, and the

710 same depth as the observations. To assess the ventilation below the mixed layer, we only used observations below 200 m.

Furthermore, we limited our assessment to observations until 2004 as CFC-11 in the atmosphere has peaked in 1994 (Bullister,

n.d.) and subducted waters since then might already re-emerge to the surface. Thus, 506000 measurements remained. As these

measurements are not equally distributed, and strongly clustered in the Northern hemisphere (Lauvset et al., 2021), we mapped

all measurements on a regular 5°x5° grid with 11 depth levels from 200 m to 6000 m that increase with depth. In each cell on

the grid the average bias was calculated. Afterwards, the volume averaged bias was calculated for the Southern hemisphere

and the North Atlantic (limited by the equator and 65°N) (Figure A6).

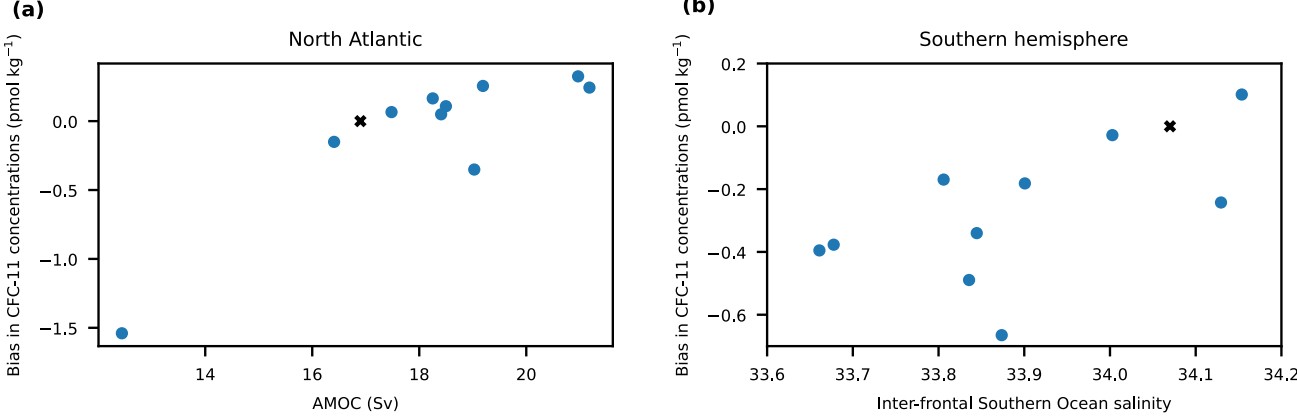

**Figure A6. Biases in subsurface CFC-11 concentrations between observations against the Atlantic Meridional Overturning circulation and the Inter-frontal Southern Ocean Salinity.** Basin-wide averaged biases in CFC-11 concentrations (observations minus simulated) below 200 m for all 10 ESMs that provided simulated CFC-11 (blue dots) **(a)** in the North Atlantic Ocean (north of the equator and limited by the Fram Strait, the Barents Sea Opening, and the Baffin Bay) and against the AMOC and **(b)** in the Southern hemisphere

(south of the equator) against the inter-frontal annual mean sea surface salinity in the Southern Ocean. The observation-based estimates for the AMOC and the inter-frontal annual mean sea surface salinity in the Southern Ocean are shown as black crosses and with zero bias in CFC-11.

### A.5 Comparison between simulated and observation-based estimates of the interior ocean $C_{ant}$ accumulation

Another way to test the here identified emergent constraint is the comparison to observation-based estimates of the interior ocean $C_{ant}$ accumulation. Here, we compare model results against the estimate for interior ocean $C_{ant}$ accumulation from 1800 to 1994 (Sabine et al., 2004) and from 1994 to 2007 (Gruber et al., 2019a), although different reconstruction methods yield different results (e.g., Khatiwala et al., 2013, their Fig. 4). While a good representation of the interior ocean $C_{ant}$ distribution is not necessarily related to a correct estimate of the air-sea $C_{ant}$ flux, it can provide an indication of the model performances and the robustness of the applied corrections. For both comparisons, we compare the multi-model mean and standard deviation and results from the ESM that represents best the three observational predictors (i.e., GFDL-ESM4). GFDL-ESM4 has a global ocean Revelle factor of 10.37, an inter-frontal sea surface salinity of 34.00, and an AMOC of 18.25. The biases that may exist in the multi-model mean, such as too little $C_{ant}$ in the Southern hemisphere due to a too low multi-model averaged sea surface salinity, should be smaller for GFDL-ESM4.

The comparison to the observation-based estimate of $C_{ant}$ accumulation from 1800 to 1994 (Sabine et al., 2004) demonstrates that the ESMs represent the distribution of $C_{ant}$ in the ocean between the basins and different latitudinal regions well (Table A5). Small underestimations exist in the Indian and Atlantic tropical ocean as well as in the southern subpolar Atlantic Ocean. The differences in the Indian Ocean may well be to observational uncertainties that are especially large in this relatively under-sampled ocean basin (Sabine et al., 2004; Gruber et al., 2019a). The underestimation in Southern Atlantic and the Atlantic sector of the Southern Ocean are consistent with an underestimation of the formation of mode and intermediate waters in the Southern Ocean due to a too low sea surface salinity. This underestimation is strongly reduced in the GFDL-ESM4 model (Table A6) indicating that the better representation of the inter-frontal sea surface salinity in the Southern Ocean also improves the simulated distribution of $C_{ant}$ in the ocean. Furthermore, GFDL-ESM4 also simulates slightly higher $C_{ant}$ in the North Atlantic, consistent with its slightly too high AMOC.

The comparison for the period from 1994 to 2007 also indicates that the ESMs on average simulate the $C_{ant}$ interior storage pattern as estimated based on observations (Gruber et al., 2019a) (Table A7). The ESMs agree with the observation-based estimates with respect to the basin and hemispheric distribution. However, they underestimate on average the storage in the Southern hemisphere in line with the underestimation of the formation of intermediate and mode waters in the Southern Ocean. When only considering GFDL-ESM4 (Table A8), this underestimation is reduced and all other regions show very good agreement.

Remaining small difference in both comparisons may be also due to different alignments of the basin boundaries, an unknown distribution of the $C_{ant}$ that entered the ocean before 1850 and has been advected 50 years longer in the ocean interior in case of Sabine et al. (2004), a different decadal variability in GFDL-ESM4 than in the real world in the case of Gruber et al. (2019a), and uncertainties in the observation-based estimates. Despite all these potential pitfalls, the 3-D repartition of $C_{ant}$ between observation-based products and ESMs agree and the model that best simulates the three key predictors, GFDL-ESM4, is almost identical to the observation-based estimates.


**Table A5. Distribution of $C_{ant}$ inventories in Pg C by basin and latitude band for 1994. The first number in each cell is the multi-model mean and standard deviation across all 18 ESMs from CMIP6 and the second number is from Table S1 in Sabine et al. (2004).**

|  | Atlantic | Pacific | Indian | World |
|---|---|---|---|---|
| 50-65°N | 4±1 / 4 | 1±0 / 1 | / | 5±1 / 5 |
| 14-50°N | 14±3 / 16 | 11±1 / 11 | 1±0 / 1 | 27±3 / 28 |
| 14°S-14°N | 4±1 / 7 | 9±2 / 8 | 4±1 / 6 | 17±3 / 21 |
| 14-50°S | 8±2 / 11 | 17±3 / 18 | 15±2 /13 | 39±6 / 42 |
| >50°S | 3±1 / 2 | 6±1 / 6 | 3±1 / 2 | 11±3 / 10 |
| total | 33±6 / 40 | 43±5 / 44 | 22±3 / 22 | 102±13 / 106 |


**Table A6. Distribution of $C_{ant}$ inventories in Pg C by basin and latitude band for 1994. The first number in each cell are derived from GFDL-ESM4 and the second number is from Table S1 in Sabine et al. (2004).**

|  | Atlantic | Pacific | Indian | World |
|---|---|---|---|---|
| 50-65°N | 6 / 4 | 1 / 1 | / | 7 / 5 |
| 14-50°N | 18 / 16 | 12 / 11 | 1 / 1 | 31 / 28 |
| 14°S-14°N | 5 / 7 | 11 / 8 | 5 / 6 | 21 / 21 |
| 14-50°S | 9 / 11 | 20 / 18 | 15 /13 | 44 / 42 |
| >50°S | 5 / 2 | 6 / 6 | 3 / 2 | 14 / 10 |
| total | 45 / 40 | 49 / 44 | 23 / 22 | 117 / 106 |

**Table A7. Distribution of $C_{ant}$ inventories in Pg C by basin and hemisphere from 1994 to 2007. T he first number in each cell is the multi-model mean and standard deviation across all 18 ESMs from CMIP6 and the second number is from Table 1 in Gruber et al. (2019).**

|  | Atlantic | Pacific | Indian | Other basins | Global |
|---|---|---|---|---|---|
| Northern hemisphere | 6.7±1.0 / 6.0±0.4 | 5.0±1.0 / 5.2±0.6 | 0.7±0.4 / 0.8±0.4 | 1.1±0.3 / 1.5±0.6 | 13.4±1.8 / 13.5±1.0 |
| Southern hemisphere | 3.5±1.0 / 5.9±1.2 | 7.4±1.0 / 8.0±1.2 | 5.6±1.3 / 6.3±3.4 | / | 16.5±2.1 / 20.1±3.8 |
| Entire basin | 10.1±1.5 / 11.9±1.3 | 12±1 / 13.2±1.3 | 6.3±1.5 / 7.1±3.4 | 1.1±0.3 / 1.5±0.6 | 29.9±3.2 / 33.7±4.0 |


**Table A8. Distribution of $C_{ant}$ inventories in Pg C by basin and hemisphere from 1994 to 2007. The first number in each cell are derived from GFDL-ESM4 and the second number is from Table 1 in Gruber et al. (2019).**

|  | Atlantic | Pacific | Indian | Other basins | Global |
|---|---|---|---|---|---|
| Northern hemisphere | 6.6 / 6.0±0.4 | 5.1 / 5.2±0.6 | 0.9 / 0.8±0.4 | 1.6 /1.5±0.6 | 14.2 / 13.5±1.0 |
| Southern hemisphere | 4.6 / 5.9±1.2 | 7.9 / 8.0±1.2 | 7.7 / 6.3±3.4 | / | 20.2 / 20.1±3.8 |
| Entire basin | 11.2 / 11.9±1.3 | 13±0 / 13.2±1.3 | 8.6 / 7.1±3.4 | 1.6 / 1.5±0.6 | 34.4 / 33.7±4.0 |



**Code availability**

The mocsy2.0 code is publicly available via https://github.com/jamesorr/mocsy.

**Data availability**

All model output from CMIP is available via https://esgf-node.llnl.gov/search/cmip6/.

**Author Contributions**

Conceptualization: JT

Methodology: JT

Software: JT

Investigation: JT

Visualization: JT

Funding acquisition: TLF, FJ

Project administration: TLF, FJ

Writing – original draft: JT

Writing – review & editing: JT, TLF, FJ

**Competing interests**

Authors declare that they have no conflict of interests.


**Acknowledgements**

This work was funded by the European Union's Horizon 2020 research and innovation programme under grant agreement No

821003 (project 4C, Climate-Carbon Interactions in the Current Century) (JT, TLF,FJ) and No 820989 (project COMFORT,

Our common future ocean in the Earth system-quantifying coupled cycles of carbon, oxygen and nutrients for determining and

achieving safe operating spaces with respect to tipping points) (TLF,FJ), and by the Swiss National Science Foundation under

grant PP00P2_198897 (TLF) and grant #200020_200511 (JT, FJ). The work reflects only the authors' view; the European Commission and their executive agency are not responsible for any use that may be made of the information the work contains. We also thank Donat Hess for his work on the North Atlantic anthropogenic carbon uptake during his master thesis at our institute, as well as Friedrich Burger, Nadine Goris and Jens Müller for discussions. We also thank one anonymous reviewer and Roland Séférian for their careful and helpful assessment of our manuscript.

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
