# Peer review of "Observation-constrained estimates of the global ocean carbon sink from Earth System Models"

_Biogeosciences, 2022_

## Author Comment (AC1)

**Response to the reviewers**

We thank the two reviewers for their careful assessment of our work. We address the reviewers' comments point by point. Added or revised text is given in blue fonts, the line numbers refer to the previously submitted manuscript without track changes. For convenience, we added a revised version of the manuscript with the planned changes highlighted after our reply to the Community Comment.

**Reviewer #1**

Review of "Observation-constrained estimates of the global ocean carbon sink from Earth system model"

The study applies observational constraints to adjust Earth system model estimates of the global ocean carbon sink. The observational constraints are the sea surface salinity in the subtropical-polar front in the Southern Ocean (as applied previously by the authors in Terhaar et al, 2021), the Atlantic Meridional Overturning Circulation (AMOC) and the Revelle buffer factor. These observational choices are plausible and the benefits of applying them are clearly set out in an iterative manner in Figure 3. The outcome is a slight elevation of the global ocean carbon sink and almost a halving of the model uncertainty, which are important improvements.

**Thank you.**

The study is comprehensive and written up in a detailed manner. In places the level of detail seemed to detract from the central message, such as discussing the details of the biological contributions when that appears to be a rather minor contribution in the global carbon uptake for anthropogenic timescales.

As suggested by the reviewer, the details about the biological contribution were removed from the text. Only one sentence was kept highlighting the minor contribution.

The only concern I raise is the particular choice of the observational constraints and while this set of choices is plausible, there are other choices that might have led to similar improvements.

So including a discussion of other choices would be helpful to the reader. For example, would a measure of the strength of the winds at key locations provide a similar benefit to the measure of the AMOC or a measure of the winter mixed layer thickness derived from Argo be beneficial? The AMOC might be used here as a proxy for ocean ventilation, but that need not be the case with gyre-scale subduction not being causally related to the AMOC. The use of the Revelle buffer factor is a plausible constraint, but the justification for that could be expanded, see possible theoretical links that can be explored or are there other references that can be utilised?

We have extended the introduction and discussion of the Revelle buffer factor as well as the references and now better describe the role of the Revelle factor as a plausible constraint:

"While the circulation determines the volume that is transported into the deeper ocean, the Revelle factor (Revelle and Suess, 1957; Sabine et al., 2004) determines the concentration of  $C_{ant}$ in these water masses. The Revelle factor describes the biogeochemical capacity of the ocean to take up  $C_{ant}$ . This biogeochemical capacity is strongly dependent on the amount of carbonate ions in the ocean that react with  $CO_2$  and  $H_2O$  to form bicarbonate ions (Egleston et al., 2010; Goodwin et al., 2009; Revelle and Suess, 1957). The more  $CO_2$  is transferred via this reaction to bicarbonate ions, the more can be taken up again from the atmosphere. The available amount of carbonate ions for this reaction depends sensitively on the difference between ocean alkalinity and dissolved inorganic carbon ( $C_1$ ) (Figure A.1.2) (Egleston et al., 2010; Goodwin et al., 2009; Revelle and Suess, 1957), highlighting the importance of alkalinity for the global ocean carbon uptake (Middelburg et al., 2020). As the buffer factor influences the  $C_{ant}$  uptake, it also exerts a strong control on the transient climate response, i.e., the warming per cumulative  $CO_2$  emissions (Katavouta et al., 2018; Rodgers et al., 2020)."

We also extended the Discussion in the Revelle factor as a predictor:

"Eventually, we have also tested the robustness of the biogeochemical predictor, by varying the definition of the Revelle factor. First, the Revelle factor was only calculated north of  $45^{\circ}$ N and south of  $45^{\circ}$ S, assuming that the high-latitude regions are responsible for the largest  $C_{ant}$  uptake, and second, the global Revelle factor was calculated by weighting the Revelle factor in each cell by the multi-model mean cumulative  $C_{ant}$  uptake from 1850 to 2100 in that cell so that the Revelle factor in cells with larger uptake is more strongly weighted. Under both definitions, the results remain almost unchanged (Table A.1.4). Furthermore, the Revelle factor has been shown here to improve the  $C_{ant}$  uptake in the Atlantic and Southern Ocean and has been earlier shown to determine the  $C_{ant}$  uptake in the tropical Pacific Ocean (Vaittinada Ayar et al., 2022), suggesting that the Revelle factor is a robust predictor of global and regional ocean  $C_{ant}$  uptake."

Furthermore, other observational constraints for the circulation might indeed provide similar improvements. For emergent constraints, it is important to find the best compromise between the best linear relationship and between the best observable predictor variable. Even a perfect emergent relationship  $(r^2=1)$  does not help to reduce uncertainties if the predictor variable cannot be observed accurately.

In the Southern Ocean, we have chosen the inter-frontal surface salinity as the best compromise. Although the surface density or even the volume of ventilated waters or a stratification index (Bourgeois et al. (2022)) may provide a more direct relationship to mode and intermediate water formation, the salinity is a good proxy (as demonstrated in Terhaar et al. (2021)) and easier to observe than the sea surface density or ocean interior variables which usually come with larger uncertainties than sea surface salinity. We now write:

"For the Southern Ocean, the verification of the link between sea surface salinity and  $C_{ant}$ uptake was previously done by linking the sea surface salinity, to the density, and to the volume of intermediate and mode waters in each model. Furthermore, the robustness of the constraint was tested against changes in the definition of the inter-frontal zone (Terhaar et al., 2021). In addition, other potential predictors were tested, such as the magnitude and seasonal cycle of sea-ice extent, wind curl, and the mixed layer depth, and upwelling strength of circumpolar deep waters. All these variables are known to influence air-sea gas exchange, freshwater fluxes, and circulation and, in turn, salinity and  $C_{ant}$ uptake. However, none of these factors alone explains biases in the surface salinity and  $C_{ant}$  uptake in the Southern Ocean. Therefore, the sea surface salinity that emerges as a result of all these individual processes represents, so far, the best variable in terms of mechanistic explanation and observational uncertainty to bias-correct models for Southern Ocean  $C_{ant}$  uptake. Further evidence for the underlying mechanism of the relationship between Southern Ocean sea surface salinity and  $C_{ant}$  uptake was provided by a later study that analysed explicitly the stratification in the water column (Bourgeois et al., 2022). Here, we further showed that the Southern Ocean  $C_{ant}$  uptake constrained by the Revelle factor and the inter-frontal sea surface salinity compares much better to observation-based estimates than the unconstrained estimate, further corroborating the identified regional constraint and mechanism (section 3.2.1)."

Along the same lines, we explored other predictors in the North Atlantic. One of these, the area where the water column is weakly stratified, was also included in the present manuscript (Table A.1.4) and which yields similar results. We had also investigated the mixed layer depth, which turned out to be biased in the ESMs by very deep mixed layers up to 5000 m so that the observed and simulated mixed layers are not comparable. Furthermore, Goris et al. (2018) also showed that the relative amount of  $C_{ant}$  that is stored below 1000 m can be used as a constraint. However, the observation-based estimates of the relative amount of  $C_{ant}$  have large uncertainties. In the end, we have chosen the AMOC because it eventually determines the amount of water that is transported southward and hence from the surface to the deep below 1000 m and hence comprises indications from different relationships. Moreover, it is relatively well observed and hence provides a good observable constraint.

In summary, this is a comprehensive study that provides a plausible adjustment of Earth system model output to improve their projections of the global ocean carbon sink. I think that this work is important and I recommend acceptance subject to the minor points raised being addressed.

Thank you.

**Detailed points;**

L47 The text is assuming that the AMOC is leading to the basin-scale subduction. I think that this statement is combining together two different processes. Subduction in ocean basins is primarily linked to the gyre circulation and the vertical and lateral transfer from the winter mixed layer to the thermocline. The AMOC is a longitudinally-averaged overturning circulation that contributes to the ventilation process by redistributing heat and tracers, but is not the same as subduction.

**Changed from subducted to ventilated as suggested by the reviewer.**

L50 The Revelle factor certainly does affect the capacity of the ocean to take up carbon. This aspect could be expanded more. The air-sea partitioning of carbon is affected by the buffer factor (Goodwin et al., 20008 & 2009; Katavouta et al., 2018). In addition, the air-sea equilibration timescale, tau, for carbon dioxide is affected by the buffer factor, tau =( $h/K_g$ )(DIC/(B CO2) where h is mixed layer thickness,  $K_g$  is exchange velocity, DIC is dissolved inorganic carbon, B is the buffer factor and CO2 is dissolved CO2.

As suggested by the reviewer, the sentences about the Revelle factor were extended to an independent paragraph (see response above).

L106 Improve syntax,"so-estimated" L109 Improve wording L163 Adjust wording

All Changed.

L173 An important point is being made as the role of the salinity and AMOC in determining water-mass formation. A list of 4 references are provided, but are they being cited as to their work on water-mass formation or did they propose the connections tbetween salinity and the AMOC to water-mass formation?

The references were provided because they state the importance of these regions for water-mass formation. Thanks to the reviewer, we realized that they were badly placed and replaced them with references that directly address the link between the salinity, the AMOC, and the water-mass formation (Goris et al., 2018, 2022; Terhaar et al., 2021).

Figure 3 is very clear and key to the study.

Thank you.

L230-231. Perhaps reword to make clearer.

The sentence was cut in two sentences to make it clearer.

L251 Cut hence.

Changed as suggested.

L297 Buckley and Marshall provided a review of heat transport, but did they make the point about anthropogenic carbon uptake?

The reference was changed to Winton et al. (2013)..

Appendix A3 Equation (2) and perhaps (3) are central to the study. I would recommend that this subsection moved into the heart of the paper.

As suggested by the reviewer, the entire appendix A3 was moved into the main part of the paper.

References

Goodwin, P., R.G. Williams, A. Ridgwell and M.J. Follows, 2009. Climate sensitivity to the carbon cycle modulated by past and future changes in ocean chemistry. Nature Geosciences, doi:10.1038/ngeo416

Goodwin, P., M.J. Follows and R.G. Williams, 2008. Analytical relationships between atmospheric carbon dioxide, carbon emissions and ocean processes. Global Biogeochemical Cycles, 22, GB3030, doi:10.1029/2008GB003184

Katavouta, A., R.G. Williams, P. Goodwin and V. Roussenov, 2018. Reconciling atmospheric and oceanic views of the Transient Climate Response to Emissions. Geophysical Research Letters, 45, 6205-6214, doi.org/10.1029/2018GL077849

**Reviewer #2 – Roland Séférian**

In this manuscript, Terhars et al. investigate how Earth system models estimates of the global ocean carbon sink can be constrained by a combination of physical parameters (the sea-surface salinity and the strength of the Atlantic Meridional Overturning Circulation) and a biogeochemical parameter (the Revelle factor).

The manuscript is timely, clearly written and proposes a sound methodology. The results are well explained and discussed through the manuscript. This work presents an important basis for the research community studying the ocean carbon cycle as this work proposes a first approach to bring together estimates of ocean carbon sink based on observational data with those based on Earth system models' simulations. I liked very much the fact that the authors explain step by step the use of a suite of emergent constraints and then perform several validations to test the robustness of their approach.

**Thank you.**

I only have one major comment and a set of minor comments/suggestions that aims to clarify some point of the paper.

**Major comments:**

Although the authors did a great job in defining and applying observational constraints to improve Earth system models' simulations/projections, they miss to thoroughly discuss how each physical or biological parameters are correlated between each other. For instance, pattern of sea-surface salinity is linked to water mass properties, which is in turn, tightly linked to large-scale circulation (deacon cells and the strength of the AMOC). Same caveat could hold for the buffer factor (globally average) which result from biological but also from chemical properties of the models.

The regression was done in one step determining the three coefficients (slope) together. Therefore, a possible, but non-existent, correlation between the different parameters is accounted for (see response directly below).

If constraining fields are correlated between each other in the observations and/or in the ESMs, this might bring light on a more mechanistic explanation of the "cascade of errors" = hydrodynamics => large-scale circulation => buffer factor rather than a "sum of errors" = hydrodynamics + large-scale circulation + buffer factor. This might be needed as a justification of applying this set of observational constraints (avoid cherry picking).

We have tested this hypothesis. The correlation coefficient  $(r^2)$  between the AMOC and the interfrontal sea surface salinity is 0.03, the one between the AMOC and the Revelle factor is 0.00, and the one between the Revelle factor and the sea surface salinity is 0.10. The correlation is in no case significant (p

Figure R.1 Surface ocean Revelle factor against the surface alkalinity and dissolved inorganic carbon. Basin-wide averaged surface ocean Revelle factor as simulated by 18 ESMs from CMIP6 against the basin-wide averaged surface ocean  $C_T$  (left), and  $A_T$  (right). The observation-based estimates from GLODAPv2 are shown as black crosses. The Revelle factor in each ESM was adjusted for biases in the surface ocean  $C_T$  (see Appendix A.1). The blue dots indicate models without calibration and/or tuning as in table 3 from Séférian et al. (2020).

Finally, in the light of deficiency/weakness of observational-based estimates of the ocean carbon sink, it might be interesting to decompose your approach on regional/basin scale uptake. Driving mechanisms, long-term trends and variability of the North Atlantic carbon sink is better understood than those of the Southern Ocean (which suffer from incomplete observational mapping across seasons). As such, does the model (and your observational constraints) help to improve the agreement between model and observation-based estimates. Besides, does the ratio in carbon uptake in the North Atlantic and the Southern Ocean is well captured between models. In the context of this paper, I wonder how far this ratio might be an additional constraint to test or a verification measure to assess the robustness of your approach.

As suggested by the reviewer, we have decomposed our approach on a regional scale by analysing the North Atlantic and Southern Ocean separately. In each case, we have used the respective circulation constraint (AMOC for the North Atlantic, inter-frontal sea surface salinity for the Southern Ocean) and the basin-wide averaged surface ocean Revelle factor. The regional fluxes were adjusted for the late starting date in the same way as the global fluxes were adjusted. The 12 Pg C that were estimated to have entered the ocean before 1850 (Bronselaer et al., 2017) were divided according to the relative uptake of each region in the multi-model mean from 1850 to 2005, i.e., 42% (5.1 Pg C) in the Southern Ocean and 15% (1.8 Pg C) in the North Atlantic.

For the Southern Ocean, we have added the following section to the manuscript:

**"3.2.1 Southern Ocean**

While the constraints were applied globally, they can also be applicable regionally as shown for the inter-frontal sea surface salinity in the Southern Ocean (Terhaar et al., 2021). Here, we update the regional constraint in the Southern Ocean with the now additionally available ESMs and extent the constraint by adding the basin-wide averaged Revelle factor in the Southern Ocean as a second variable. For the period from 1765 to 2005, the simulated multi-model mean air-sea Cant flux that is adjusted for the late starting date is  $63.5 \pm 6.1 Pg C$ . Please note that the numbers here are for fluxes from 1765 to 2005 and are not the same as in Terhaar et al. (2021b), where fluxes from 1850 to 2005 were reported. The two-dimensional constraint shows a higher correlation coefficient ( $r^2=0.70$ ) than the one-dimensional constraint when only the inter-frontal sea surface salinity is used as a predictor ( $r^2=0.62$ ). Slight differences to Terhaar et al. (2021b) exist due to the additional ESMs that are by now available. When exploiting this relationship with observations of the Southern Ocean Revelle factor  $(12.19\pm0.01)$  and the sea surface salinity, the best estimate of the cumulative air-sea Cant flux from 1765 to 2005 in the Southern Ocean increases to 72.0±3.4 Pg C. In comparison, observation-based estimates for the same period report 69.6±12.4 Pg C (Mikaloff Fletcher et al., 2006) and 72.1±12.6 Pg C (Gerber et al., 2009). The constrained thus reduces the uncertainty not only globally but also in the Southern Ocean by 44%."

For the Atlantic Ocean, we now write:

**"3.2.2 Atlantic Ocean**

As for the Southern Ocean, we also apply a two-dimensional constraint to the Atlantic Ocean, using the AMOC and the basin-wide averaged surface ocean Revelle factor in the North Atlantic as predictor. The unconstrained cumulative air-sea  $C_{ant}$  flux from 1765 to 2005 in the North Atlantic adjusted for the late starting date is  $21.9 \pm 3.3$  Pg C. For this period, the two-dimensional constraint results in a relationship with a correlation coefficient of 0.57. If only the AMOC had been used the correlation factor would have been 0.49. When exploiting this relationship with observations of the North Atlantic Revelle factor and AMOC, the best estimate of the cumulative air-sea  $C_{ant}$  flux from 1765 to 2005 in the Atlantic Ocean increases to  $22.7\pm2.2$  Pg C. In comparison, observation-based estimates are  $20.4\pm4.9$  Pg C (Mikaloff Fletcher et al., 2006) and  $20.4\pm6.5$  Pg C (Gerber et al., 2009). The constrained and unconstrained estimates are both above the observation-based estimates but within the uncertainties. The constrained estimate is even higher than the unconstrained one, but only by 0.8 Pg C, and its uncertainty is reduced by 33%."

As suggested by the reviewer, we have also analyzed the ratio of the  $C_{ant}$  uptake in the Atlantic and Southern Ocean (Fig R2). No significant relationship can be found (r2=0.05, p=0.38), although the models tend to underestimate the ratio albeit with a large uncertainty. The constrained estimate of the ratio is close to the observation-based estimate, giving further confidence to the robustness of our estimates. No changes are made to the manuscript.

**Figure R.2 Cumulative global ocean anthropogenic carbon uptake against the ratio of uptake in the Atlantic and Southern Ocean.** The cumulative global uptake was calculated from 1765 to 2010. The ratio was calculated from 1765 to 2005, the same period over which the data-based fluxes were available. The mean is shown for Mikaloff-Fletcher et al. (2006) (black dashed line), for Gerber et al. (2008) (magenta dashed line) and for the here derived constrained local estimates from section 3.2.1 and 3.2.2 in the revised manuscript (red dashed line). The uncertainty is only shown for Mikaloff-Fletcher et al. (2006), accounting for uncertainties in transport, and would even be larger for Gerber et al. (2009), accounting for both transport and Cant reconstruction uncertainties.

Regarding the conclusions of the paper, I think the authors could make a stronger point resulting from this work. First, I might be relevant to discuss the consequence of this work on the carbon budget (Friedlingstein et al. 2022), especially in the context of the budget imbalance term. Revised (contrained) estimates appears to be about 10% higher than the unconstrained estimates. The magnitude of the revision is thus greater than the budget imbalance. What would be the consequence then? a weaker land-surface carbon sink?

Here, we have only addressed the budget imbalance for the historical period from 1850 to 2020. Over such a long time period (i.e. 171 years), different phasing in simulated decadal and interannual variabilities as simulated by fully coupled Earth system models average out and Earth System Models can be well compared to observation-based estimates. We find that the ocean sink was underestimated, and this adjustment of the ocean sink accounts for roughly two thirds of the budget imbalance. The remaining one third would still have to be explained otherwise. We thank the reviewer for supporting this point. Although this was clearly indicated in the Results section, we have now also precised this in the Conclusion:

"The here provided improved estimate of the size of the global ocean carbon sink may help to close the carbon budget imbalance since 1850 (Friedlingstein et al., 2022)"

For the historical period, our results suggest that the ESMs from CMIP6 underestimate the longterm uptake. Due to the different phasing of simulated unforced interannual-to-decadal variabilities in the ESMs, the unforced, internal decadal variability over the last decades cannot be assessed by ESMs. However, the ESMs allow also to quantify the long-term mean flux (e.g., multi-decadal) from increasing atmospheric carbon and climate change. Given that the here quantified CMIP based ESM estimates agree with the observation-based that were used in the Global Carbon Budget 2021 over the entire period from 1990 to 2020 and that both estimates are larger than the hindcast models that were used in the Global Carbon Budget 2021, it is highly unlikely that natural variability sustains in a particular phase for a 30-yr period in the ESMs. Our results suggest that the hindcast models in the Global Carbon Budget 2021 indeed underestimate the ocean carbon sink. Unfortunately, the hindcast model output of the Global Carbon Budget models is not openly accessible, and we thus cannot not make an analysis of the different predictors to explain this difference. Therefore, we are left with recommending that such an analysis should be done.

As suggested by the reviewer we have extended the Conclusion along these lines:

"Moreover, biases in these quantities and corrections for the late starting date may well be the reason for offset between models and observations over the last 30 years (Hauck et al., 2020; Friedlingstein et al., 2022). Although the here identified constraints cannot correct for misrepresentation of the unforced decadal variability, such variability plays likely a minor role when averaging results over longer periods. Indeed, we find good agreement between our estimate and the observation-based estimate from the Global Carbon Budget 2021 for the period from 1990 to 2020. This agreement suggests that the hindcast models underestimate the ocean  $C_{ant}$  uptake. This underestimation is thus likely the explanation for the difference between models and observation-based product in the Global Carbon Budget (Friedlingstein et al., 2022). However, the output of the Global Carbon Budget hindcast models is not publicly available for evaluating possible data-model differences for the inter-frontal sea surface salinity, the AMOC, and the Revelle factor."

We also added a sentence to the abstract:

"Our constrained results are in good agreement with the air-sea  $C_{ant}$  estimates over the last three decades based on observations of the  $CO_2$  partial pressure at the ocean surface in the Global Carbon Budget 2021, and suggest that existing hindcast ocean-only model simulations underestimate the global ocean anthropogenic carbon sink."

On the other hand, in a context of improving estimates of the carbon feedbacks, what would be the revision of the Beta and Gamma as inferred from your approach. In might be interesting to include in your work ssp585-bgc (which has been conducted by most of the modelling center) and see how your approach works on ocean Beta and Gamma.

This is indeed an interesting idea. However, as far as we can find the model output on https://esgfnode.llnl.gov/search/cmip6/, only 7 ESMs have provided the ssp585-bgc runs. It would not be statistically robust to fit a 3-dinemsional multi-linear regression based on 7 data points. Furthermore, uncertainties exist with respect to beta and gamma in the SSP runs due to non- $CO_2$  radiative forcing and land-use change that are not existing in the idealized 1% runs, which are usually used to calculate beta and gamma (e.g. Arora et al. 2020). However, in the 1%, there is no historical period over which the predictors in the models could be identified. Therefore, we decided not to include such an analysis and leave it to subsequent studies to explore this further.

**Minor comments:**

**L12: explain the buffer factor in the abstract**

As suggested by the reviewer, we have added the following sentence to the abstract:

"The Revelle factor quantifies the chemical capacity of seawater to take up carbon for a given increase in atmospheric CO2."

Figure 1: please use the same temporal baseline for panel a) and b). from 1950 onwards?

The x-axis in Figure 1a was chosen over the period where data exists for all estimates, the CMIP6 models and observation-based and hindcast model estimates from the Global Carbon Budget 2021. We prefer to keep the zoom to make all these estimates comparable. In Figure 1b, we have chosen 1950, so that the longer-term perspective can be seen. We keep the manuscript unchanged.

L106 Improve syntax,"so-estimated"

Changed.

L157: Many other papers have used emergent/observational constraints (Boé et al., Bourgeois et al., Cox et al., Douville et al., Plazzotta et al., Schlund et al., etc....) — They can also be listed here.

We have added Bourgeois et al. (2022) because it also focuses on the ocean carbon sink and is already part of the reference list. The other references could be added but the reference list is already long. If the reviewer wants us to add another reference for a special reason, we can do that at any time.

Figure 2: please add 'the strength of' before "the Atlantic meridional..."

Changed as suggested.

Figure 3: Please add R-square for each panels c, e and g as an indication of the quality of the fit

It is not possible to add R-squared for each panel, as all three predictors are fitted at the same time. However, we can calculate the difference in  $r^2$  that each predictor makes when added last. For example, the air-sea CO2 flux from 1994 to 2007 has an  $r^2$  of 0.87. If the Revelle factor is removed,  $r^2$  decreases to 0.75, if the AMOC is removed  $r^2$  decreases to 0.64, and if the interfrontal salinity is removed  $r^2$  decreases to 0.59. Thus, the salinity seems to have the largest impact, the AMOC the second largest and the Revelle factor the least impact. To highlight the simultaneous fit, we followed the recommendation by both reviewers to bring the core equations into the main part of the manuscript.

On this figure, it is unclear if model estimate are based on multiple realisation or just one single member

For clarification, we have added the following sentence to the figure legend:

"For each ESM, one ensemble member was used as the difference between ensemble members has been shown to be small compared to the inter-model differences (Terhaar et al., 2020, 2021)."

L237: one can also consider the CO2 mole fraction that is \*really seen\* by the ocean carbon module because of various treatment of the air-sea CO2 exchange (Hauck et al. 2020, already mentioned in this work)

As suggested by the reviewer, we have added the following words to the sentence:

"...neglecting the water vapour pressure when calculating the local pCO2 in each ocean grid cell (Hauck et al., 2020) as is done in CMIP models (Orr et al., 2017),..."

L341: Conclusion — see above comments

The Conclusion was changed according to the above-mentioned comments.

Appendix: Biogeosciences allows more materials than short/letter paper, I would recommend to move some of the material of the appendix into the heart of the paper. Some of them are central to your work.

We have moved some parts into the main part, as suggested by the reviewer.

Table A.1.1 please consider adding data citation doi (where relevant) for improving the reproducibility of the work.

The table is already large, and we prefer not to overload it. However, we have no strong opinion and can add the doi if the editor and reviewer prefers to.

References:

Boé, J., Hall, A. & Qu, X. September sea-ice cover in the Arctic Ocean projected to vanish by 2100. Nat. Geosci. 2, 341–343 (2009).

Bourgeois, T., Goris, N., Schwinger, J. *et al.* Stratification constrains future heat and carbon uptake in the Southern Ocean between 30°S and 55°S. *Nat Commun* 13, 340 (2022). https://doi.org/10.1038/s41467-022-27979-5

Cox, P., Pearson, D., Booth, B. *et al.* Sensitivity of tropical carbon to climate change constrained by carbon dioxide variability. *Nature* 494, 341–344 (2013). https://doi.org/10.1038/nature11882

Douville H., M. Plazzotta (2017) Midlatitude summer drying : An underestimated threat in CMIP5 models ? Geophys. Res. Lett., 44, 9967-9975, doi:10.1002/2017GL075353

Hauck, J., Zeising, M., Le Quéré, C., Gruber, N., Bakker, D. C. E., Bopp, L., Chau, T. T. T., Gürses, Ö., Ilyina, T., Landschützer, P., Lenton, A., Resplandy, L., Rödenbeck, C., Schwinger, J., Séférian R.:Consistency and challenges in the ocean carbon sink estimate for the Global Carbon Budget. Front. Mar. Sci., 7, 852. https://www.frontiersin.org/article/10.3389/fmars.2020.571720, 2020.

Plazzotta, M. et al. : Land surface cooling induced by sulfate geoengineering constrained by major volcanic eruptions. Geophysical Research Letters, 45, 5663–5671. https://doi.org/10.1029/2018GL077583, 2018.

Schlund, M., Lauer, A., Gentine, P., Sherwood, S. C., and Eyring, V.: Emergent constraints on equilibrium climate sensitivity in CMIP5: do they hold for CMIP6?, Earth Syst. Dynam., 11, 1233–1258, https://doi.org/10.5194/esd-11-1233-2020, 2020.

**Comment by Nicolas Gruber**

The first criticism is that the Earth System Models (ESM) applied are structurally biased and therefore not suited for the task because they are not eddy-resolving. If taken seriously and indeed true, this would dismiss almost the entire modeling literature, including many of the commentator's studies. We add a discussion on results from eddy-resolving models, drawing from our earlier analysis (Terhaar et al, 2021).

The second issue raised is the role of interannual-to-decadal climate variability of global or basin-scale air-sea carbon fluxes. We agree that the phasing of internal, unforced variability in fully coupled ESMs is by design not in line with observation-based estimates. Also, the magnitude of this variability maybe biased. However, this variability is largely irrelevant on century time scales and thus for large parts of our study and main results, such as the entire historical period and the next hundred years.

We now also present an air-sea carbon flux estimate for the 31-year period from 1990 to 2020, likely long enough to largely avoid potential biases from interannual-to-decadal climate variability. Over this longer period, the constrained ocean  $C_{ant}$  sink based on ESMs is in excellent agreement with the surface ocean observation-based estimates of the ocean  $C_{ant}$  sink and significantly larger than the hindcast models in the Global Carbon Budget.

Finally, the commentator asks for a comparison of model results with reconstructions of ocean anthropogenic carbon. We now provide such a comparison. We also include a comparison with air-sea flux results from two ocean inversion studies. A comparison with CFC-11 data was already included. All these comparisons support our conclusions.

A detailed point-by-point response is given below.

**Assessment:**

Terhaar et al. use an emergent constraint approach to make essentially two arguments: Current ocean  $CO_2$  uptake estimates are 9-11% too low, and that their constraints permit them to reduce the present and past CO2 uptake by 42-59%. The topic is relevant, the method is sound, the paper is overall well written (with some exceptions), and the results are important. Thus, this study clearly deserves to be published.

But I have two important concerns that need to be addressed, in my opinion, before I can endorse the publication of this manuscript.

Robustness: In my opinion, the major conclusions, particularly the latter regarding the substantial uncertainty reduction, are not robust as presented. By using a class of non eddy-resolving models, which disregard a set of critical processes in the ocean that are known to be relevant for controlling the uptake of transient tracers through their impact on deep water formation, the results are potentially seriously biased. Thus while the results appear precise, they may not represent an accurate estimate of the global uptake.

Caveats of our study are discussed in section 5 of the manuscript. We did not discuss mesoscale eddies, as their role was analyzed in our previous study (Terhaar et al., 2021). We provide the text from Terhaar et al. (2021) here again:

"Mesoscale eddies in the Southern Ocean influence the transport of tracers, such as heat, salinity, carbon, and nutrients  $(\underline{58}-\underline{61})$ . However, the explicit simulation of these mesoscale eddies requires high horizontal and vertical ocean model resolutions, especially in high latitudes such as

the Southern Ocean (62). Most of the CMIP5 and CMIP6 models use ocean models with horizontal resolution of about 1° (22, 63). To date, conducting transient simulations with fully coupled ESMs in higher resolution is computationally too expensive, especially because these simulations also need a sufficiently long spin-up to reach a stable equilibrium (64, 65). Therefore, the effect of eddies on the mean ocean circulation and the transport of ocean tracers, such as salinity and carbon, are parametrized within the CMIP models. While the eddy parametrization has an effect on the simulated sea surface salinity and Cant uptake (58–61), this effect cannot be quantified by the state-of-the art CMIP6 ESMs due to their relatively coarse resolution and merits further investigation when eddy-resolving ocean models incorporated in global coupled ESMs will become more widely available."

We added the following text to the manuscript:

~

In addition, parametrizations of non-represented processes such as mesoscale and sub-mesoscale circulation features like small-scale eddies may lead to biases in the model ensemble. For individual models, it has been shown that changes in horizontal resolution and hence a more explicitly simulated circulation change the model physics and biogeochemistry, and hence also the ocean carbon and heat uptake (Lachkar et al., 2007, 2009; Dufour et al., 2015; Griffies et al., 2015). However, an increase in resolution does not necessarily lead to improved simulations and the changes in oceanic  $C_{ant}$  uptake maybe lower or higher, depending on the model applied. When increasing the NEMO ocean model from a non-eddying version (2° horizontal resolution) to an eddying version  $(0.5^{\circ})$ , Lachkar et al. (2009) find a decrease in the sea surface salinity by around 0.1 at the Southern Ocean surface that brings the model further away from the observed salinity, a decrease of the volume of Antarctic intermediate water and a decrease in the Southern Ocean uptake of CFC and hence likely also of  $C_{ant}$ . This example corroborates the underlying mechanism of the emergent constraint in the Southern Ocean that higher sea surface salinity directly affects the formation of Antarctic intermediate water and the uptake of Cant. Another example can be found within the ESM ensemble of CMIP6. The MPI-ESM-1-2-HR and MPI-ESM-1-2-LR have a horizontal resolution of 0.4° and 1.5° respectively but the same underlying ocean model. The high-resolution version has an inter-frontal salinity of 33.98, a Southern Ocean surface Revelle factor of 12.82, and a Southern Ocean Cantuptake from 1850 to 2005 of 56.4 Pg C. The coarser resolution version has an inter-frontal sea surface salinity of 33.92, a Southern Ocean surface *Revelle factor of 12.89, and a Southern Ocean Cant uptake of 58.0 Pg C. These differences are* much smaller than the inter-model differences (33.66-34.15 for salinity, 12.14-13.11 for the Revelle factor, and 48.8-71.1 Pg C for the Southern Ocean Cant uptake) that result from different ocean circulation and biogeochemical models, sea ice models, and atmospheric and land biosphere models, as well as the coupling between these models. These examples show that higher resolution does not necessarily lead to better results, effects potentially the predictor and the predicted variable in the same way, and that differences in the underlying model components and spin-up and initialization strategies lead so far to much larger differences between ESMs than resolution does (Séférian et al., 2020). As long as simulations with higher resolution, which are also spun-up over hundreds of years (Séférian et al., 2016), are not yet available, and potentially important processes such as changing riverine fluxes and freshwater from land ice are not included, it remains speculative if higher resolution would lead to a reduction of inter-model uncertainty, or even a better representation of the observations. Moreover, the here-identified relationships that are based on the current understanding of physical and biogeochemical oceanography and that were tested for robustness in several ways may likely also exist across ensembles of eddy-resolving models."

The notion that all processes need to be represented to estimate  $C_{ant}$  uptake is flawed. Early boxdiffusion model (Oeschger et al., 1975), calibrated with radiocarbon, are able to estimate global ocean uptake within the error limits of observations. This class of models does not explicitly resolve ocean dynamics but produces very useful results. They have passed the test of history as they exist for 50 years and their predictions are still valid (Cubasch et al., IPCC AR5, WG1, Chapter 1, 2013). Within the physical realm, the pioneering ocean-atmosphere models of Manabe, Bryan and co-workers (e.g., S. Manabe and K. Bryan, J. Atmospheric Sciences **26** (1969): 786-89) had coarse resolution and flux correction but produced groundbreaking results. We are not willing to dismiss the usefulness of an entire class of models (ESMs CMIP5 and CMIP6 models, Global Carbon Budget, RECCAP-ocean) because they do not resolve small-scale, weather-like features. Furthermore, literally all 17 ESMs used in this study fall within the uncertainty range of the observation-based estimate of  $29 \pm 5$  Pg C by Gruber et al. (2019) before they are used to constrain the Cant uptake, indicating that they are well capable of representing the historical Cant uptake.

Eddy-resolving models represent a highly interesting scientific frontier. We are looking forward to emerging eddy resolving simulations to demonstrate that they are useful to faithfully project  $C_{ant}$  uptake on the global and multi-centennial scale, which are relevant to the anthropogenic  $CO_2$  and climate perturbation.

Observational constraints: The study is entirely based on rather indirect constraints, i.e., the salinity of parts of the Southern Ocean, the surface buffer factor, and the AMOC (in decreasing order of relevance), while there are many direct constraints that the authors have decided to disregard. This may be a valid approach to provide an independent estimate, but it then behooves the authors to demonstrate that the constrained models are actually doing better against the unused observational constraints. Particularly relevant here is the three-dimensional distribution of anthropogenic CO2 in the ocean interior. Are the models that are within the best constrained range also those models that reproduce the reconstructed distribution the best?

The statement that we disregard or decided to disregard direct constraints is not true.

*First, we compare the simulated CFC-11 concentrations (direct constraints) by ESMs to observed CFC-11 concentrations from GLODAPv2 (Appendix A.4 in the revised manuscript).*

Second, we largely discussed the possible direct constraints in section 2 and concluded:

"Overall, the difference between ocean hindcast models, observation-based  $CO_2$  flux estimates, and interior ocean  $C_{ant}$  estimates as well as the uncertainties in the climate-driven change in  $C_T$ and pre-industrial outgassing indicate that uncertainties of the past ocean  $C_{ant}$  sink remain larger than the uncertainties of these individual products (Crisp et al., 2022) and do not allow to constrain the ocean  $C_{ant}$  sink"

The uncertainty in the purely observation-based ocean carbon sink estimates, such as the one from Gruber et al. (2019) for the period from 1994 to 2007, encompasses all CMIP6 models and can hence not be used to reduce the uncertainty of the model ensemble.

We have compared our estimate of global  $C_{ant}$  uptake for the period from 1994 to 2007 with the estimate from Gruber et al. (2019) and find agreement within uncertainties. We now compare airsea  $C_{ant}$  fluxes in the Southern Ocean and the North Atlantic from two ocean inversion studies with our results as suggested by reviewer #2. We find again agreement within uncertainties of the inversions. The manuscript was adapted accordingly (please see revised manuscript or responses to reviewer #2).

Although the 3D  $C_{ant}$  distribution is not necessarily correct if the  $C_{ant}$  air-sea fluxes are improved, we have compared the  $C_{ant}$  distribution in the model that performs best (GFDL-ESM4) with respect to the 3 predictor variables (Global ocean Revelle factor of 10.37, inter-frontal sea surface salinity of 34.00, and an AMOC of 18.25) with Sabine et al. (2004) and Gruber et al. (2019). For the comparison, we have scaled the interior ocean  $C_{ant}$  with a correction factor as in the manuscript but with respect to 1800 as Sabine et al. (2004) quantify changes in ocean  $C_{ant}$ since 1800 (Tables R1-R4, and Tables A.5.1-A.5.4 in the revised manuscript). Please note that different methods to reconstruct  $C_{ant}$  yield different estimates as for example illustrated by Fig. 4 in Kathiwala et al. (2013). Therefore, our comparison should be viewed with caution.

Table R1: Distribution of  $C_{ant}$  inventories in Pg C by basin and latitude band for 1994. The first number in each cell are derived from GFDL-ESM4 and the second number is from Table S1 in Sabine et al. (2004).

|                | Atlantic | Pacific | Indian  | World     |  |
|----------------|----------|---------|---------|-----------|--|
| 50-65°N        | 6 / 4    | 1 / 1   | /       | 7 / 5     |  |
| 14-50°N        | 18 / 16  | 12 / 11 | 1 / 1   | 31/28     |  |
| 14°S-14°N      | 5 / 7    | 11 / 8  | 5/6     | 21 / 21   |  |
| 14-50°S        | 9 / 11   | 20 / 18 | 15 /13  | 44 / 42   |  |
| $>50^{\circ}S$ | 5 / 2    | 6/6     | 3 / 2   | 14 / 10   |  |
| total          | 45 / 40  | 49 / 44 | 23 / 22 | 117 / 106 |  |

Table R2: Distribution of  $C_{ant}$  inventories in Pg C by basin and latitude band for 1994. The first number in each cell is the multi-model mean and standard deviation across all 18 ESMs from CMIP6 and the second number is from Table S1 in Sabine et al. (2004).

|                | Atlantic  | Pacific   | Indian    | World                     |
|----------------|-----------|-----------|-----------|---------------------------|
| 50-65°N        | 4±1 / 4   | 1±0 / 1   | /         | 5±1 / 5                   |
| 14-50°N        | 14±3 / 16 | 11±1 / 11 | 1±0 / 1   | 27±3 / 28                 |
| 14°S-14°N      | 4±1 / 7   | 9±2 / 8   | 4±1 / 6   | 17±3 / 21                 |
| 14-50°S        | 8±2 / 11  | 17±3 / 18 | 15±2 /13  | 39 ± 6 / 42 |
| $>50^{\circ}S$ | 3±1 / 2   | 6±1 / 6   | 3±1 / 2   | 11±3 / 10                 |
| total          | 33±6 / 40 | 43±5 / 44 | 22±3 / 22 | 102±13 / 106              |

Table R3: Distribution of  $C_{ant}$  inventory change in Pg C by basin and hemisphere from 1994 to 2007. The first number in each cell are derived from GFDL-ESM4 and the second number is from Table 1 in Gruber et al. (2019).

|              | Atlantic              | Pacific               | Indian        | Other basins  | Global         |   |
|--------------|-----------------------|-----------------------|---------------|---------------|----------------|---|
| Northern     | 6.6 /                 | 5.1 /                 | 0.9 /         | 1.6 /         | 14.2 /         | _ |
| hemisphere   | 6.0±0.4               | 5.2±0.6               | $0.8{\pm}0.4$ | $1.5{\pm}0.6$ | $13.5{\pm}1.0$ |   |
|              |                       |                       |               |               |                |   |
| Southern     | 4.6 /                 | 7.9 /                 | 7.7/          | /             | 20.2 /         |   |
| hemisphere   | 5.9±1.2               | 8.0±1.2               | 6.3±3.4       |               | 20.1±3.8       |   |
| Entiro hasin | 11.27                 | 13+0 /                | 86/           | 16/           | 311/           |   |
| Entire busin | 11.27
$11.9\pm1.3$ | $13\pm07$
13.2±1.3 | 7.1±3.4       | 1.5±0.6       | 33.7±4.0       |   |
|              |                       |                       |               |               |                |   |

Table R4: Distribution of  $C_{ant}$  inventory change in Pg C by basin and hemisphere from 1994 to 2007. The first number in each cell is the multi-model mean and standard deviation across all 18 ESMs from CMIP6 and the second number is from Table 1 in Gruber et al. (2019).

|              | Atlantic   | Pacific   | Indian        | Other basins  | Global     |
|--------------|------------|-----------|---------------|---------------|------------|
| Northern     | 6.7±1.0 /  | 5.0±1.0 / | 0.7±0.4 /     | 1.1±0.3 /     | 13.4±1.8 / |
| hemisphere   | 6.0±0.4    | 5.2±0.6   | $0.8{\pm}0.4$ | 1.5±0.6       | 13.5±1.0   |
|              |            |           |               |               |            |
| Southern     | 3.5±1.0 /  | 7.4±1.0 / | 5.6±1.3 /     | /             | 16.5±2.1 / |
| hemisphere   | 5.9±1.2    | 8.0±1.2   | 6.3±3.4       |               | 20.1±3.8   |
| Entire basin | 10.1±1.5 / | 12±1 /    | 6.3±1.5 /     | 1.1±0.3 /     | 29.9±3.2 / |
|              | 11.9±1.3   | 13.2±1.3  | 7.1±3.4       | $1.5{\pm}0.6$ | 33.7±4.0   |
|              |            |           |               |               |            |

**We have added the following text to the main manuscript:**

"In addition to the evaluation with observations of CFC, the comparison of the interior ocean  $C_{ant}$  distribution demonstrates first that the ESMs on average represent the observation-based distributions within the margins of error (Tables A.5.1 and A.5.3). Only in the Southern hemisphere, the ESM average remains below, as expected due to the average ESM bias towards too low inter-frontal sea surface salinities, too little formation of mode and intermediate waters, and hence too little storage of  $C_{ant}$  in the Southern hemisphere. When using the model that represents best the three predictors, GFDL-ESM4 (Dunne et al., 2020; Stock et al., 2020), the comparison to observation-based interior ocean  $C_{ant}$  distribution becomes almost identical (Tables A.5.2 and A.5.4), suggesting that a better representation of these parameters indeed improves the simulation of  $C_{ant}$  uptake and its distribution in the ocean interior."

Furthermore, we have added the comparison as an appendix, next to the CFC evaluation:

**"A.5 Comparison between simulated and observation-based estimates of the interior ocean $C_{ant}$ accumulation**

Another way to test the here identified emergent constraint is the comparison to observationbased estimates of the interior ocean  $C_{ant}$  accumulation. Here, we compare model results against the estimate for interior ocean  $C_{ant}$  accumulation from 1800 to 1994 (Sabine et al., 2004) and from 1994 to 2007 (Gruber et al., 2019a), although different reconstruction methods yield different results (e.g., Khatiwala et al., 2013, their Fig. 4). While a good representation of the interior ocean  $C_{ant}$  distribution is not necessarily related to a correct estimate of the air-sea  $C_{ant}$ flux, it can provide an indication of the model performances and the robustness of the applied corrections. For both comparisons, we compare the multi-model mean and standard deviation and results from the ESM that represents best the three observational predictors (i.e., GFDL-ESM4). GFDL-ESM4 has a global ocean Revelle factor of 10.37, an inter-frontal sea surface salinity of 34.00, and an AMOC of 18.25. The biases that may exist in the multi-model mean, such as too little  $C_{ant}$  in the Southern hemisphere due to a too low multi-model averaged sea surface salinity, should be smaller for GFDL-ESM4.

The comparison to the observation-based estimate of  $C_{ant}$  accumulation from 1800 to 1994 (Sabine et al., 2004) demonstrates that the ESMs represent the distribution of  $C_{ant}$  in the ocean between the basins and different latitudinal regions well (Table A.5.1). Small underestimations exist in the Indian and Atlantic tropical ocean as well as in the southern subpolar Atlantic Ocean. The differences in the Indian Ocean may well be to observational uncertainties that are especially large in this relatively under-sampled ocean basin (Sabine et al., 2004; Gruber et al., 2019a). The

underestimation in Southern Atlantic and the Atlantic sector of the Southern Ocean are consistent with an underestimation of the formation of mode and intermediate waters in the Southern Ocean due to a too low sea surface salinity. This underestimation is strongly reduced in the GFDL-ESM4 model (Table A.5.2) indicating that the better representation of the inter-frontal sea surface salinity in the Southern Ocean also improves the simulated distribution of  $C_{ant}$  in the ocean. Furthermore, GFDL-ESM4 also simulates slightly higher  $C_{ant}$  in the North Atlantic, consistent with its slightly too high AMOC.

The comparison for the period from 1994 to 2007 also indicates that the ESMs on average simulate the  $C_{ant}$  interior storage pattern as estimated based on observations (Gruber et al., 2019a) (Table A.5.3). The ESMs agree with the observation-based estimates with respect to the basin and hemispheric distribution. However, they underestimate on average the storage in the Southern hemisphere in line with the underestimation of the formation of intermediate and mode waters in the Southern Ocean. When only considering GFDL-ESM4 (Table A.5.4), this underestimation is reduced and all other regions show very good agreement.

Remaining small difference in both comparisons may be also due to different alignments of the basin boundaries, an unknown distribution of the  $C_{ant}$  that entered the ocean before 1850 and has been advected 50 years longer in the ocean interior in case of Sabine et al. (2004), a different decadal variability in GFDL-ESM4 than in the real world in the case of Gruber et al. (2019a), and uncertainties in the observation-based estimates. Despite all these potential pitfalls, the 3-D repartition of  $C_{ant}$  between observation-based products and ESMs agree and the model that best simulates the three key predictors, GFDL-ESM4, is almost identical to the observation-based estimates."

In summary, I have serious concerns about the conclusion drawn here. Given the structural biases that are inherent in the models and the rather indirect nature of the constraints, the proposal of a strongly reduced uncertainty for the oceanic uptake of CO2 seems far-fetched. To me this seems like a classical case for overconfidence stemming from a limited perspective of all the issues at stake.

All three authors reject the insinuation of overconfidence and limited perspective. We find such offensive language not appropriate for a review.

Unfortunately, the author of the comment does not explain the meaning of "issues at stakes". A critical issue we have not included yet in the Conclusion is research funding and the combination of ocean data with data from the atmosphere, land biosphere, ocean sediments and remote sensing as well as modelling. Important is also to improve our understanding of Earth system variability over the last million year and beyond for better projections of the future. We added in the Conclusion section:

"Despite this step forward in the understanding of ESMs, a comprehensive research strategy that combines the measurements of important physical, biogeochemical, and biological parameters in the ocean with other data streams and modelling is needed. A comprehensive approach is necessary to improve our still incomplete understanding of the global carbon cycle and its functioning in the climate and Earth system over the past and under ongoing global warming."

Indirect constraints are seen to be robust by the community if the underlying mechanism can be explained. Sanderson et al. (2020) write in a review about emergent constraints:

"Bottom-up approaches such as the process decomposition of factors controlling carbon uptake in the Southern Ocean (Terhaar et al., 2021) or the "cloud controlling factors" for individual types of cloud feedback (Klein et al., 2017) have the potential to isolate and quantify structural assumptions in composite elements of a net response, allowing the individual assessment of constraints in each component and the isolation of ensemble structural assumptions in the associated processes.

...

ECs could play a useful role by defining reduced-space metrics that consider only those aspects of model performance that are relevant to a particular future response. Multi-metric emergent constraints may provide a useful "third way": they are less sensitive to structural errors than single-metric emergent constraints and can be targeted toward processes that may drive future responses more accurately than generic performance metrics, which do not explicitly account for the relevance of an observable to a given response (Baker and Taylor, 2016; Collier et al., 2018)."

**Recommendation:**

I recommend a major revision that revisits the uncertainties of the approach taken and the conclusions that the authors draw from their work. The power of the emergent constraint rests primarily with the future, while the relevance (and novelty) for the past and presence is much less clear. I thus strongly encourage the authors to de-emphasize the discussion of the relevance for the present (which is anyway less evident since the coupled models produce their own climate variability) and instead focus the study on what the constrained ensemble can say about the future.

We are somewhat puzzled by the statement that emergent constrains are of no or limited use for the past and the present. Unfortunately, the comment provides no example or reference that would support such a statement.

One can think of process-based emergent constraints as a bias correction. This bias must be accounted for over all years, in the past, present, and the future. The constrained ESM estimate hence gives a bias-corrected value for the  $C_{ant}$  uptake over any time.

Although decadal or inter-annual variability on the air-sea  $C_{ant}$  flux is averaged out over an ensemble of ESMs, we have never claimed that decadal variability does not impact the flux estimates from 1994 to 2007.

In the legend of Table 1, we had written:

"Uncertainties from the decadal variability on shorter timescales, e.g., for 1994-2007, are not included."

When presenting the numbers in the main text the first time, we had written:

"Thus, the mismatch between observation-based air-sea  $C_{ant}$  flux estimates from 1994 to 2007 and the here provided results may not exist over a longer period of time and be caused by a different timing and magnitude of decadal variabilities in ESMs and the real world (Landschützer et al., 2016; Gruber et al., 2019b; Bennington et al., 2022), as well as uncertainties in the observationbased products (Bushinsky et al., 2019; Gloege et al., 2021, 2022)." "However, even after correcting these hindcast simulations upwards by employing the here identified emergent constraint, their corrected estimate may remain below the CMIP-derived estimate here due to the historical decadal variations in the  $C_{ant}$  uptake that is not represented with the same phasing in fully coupled ESMs (Landschützer et al., 2016; Gruber et al., 2019b; Bennington et al., 2022)."

To underline that point, we have added a constraint estimate for the entire period over which the Global Carbon Budget 2021 provides observation-based estimates of the air-sea  $C_{ant}$  flux and added the following lines to the paragraph above:

"Indeed, when the entire period for which observation-based air-sea  $C_{ant}$  flux estimates from the Global Carbon Budget are available (1990-2020), the constrained estimate of the ocean  $C_{ant}$  sink based on ESMs (80.7 ± 2.5 Pg C) is very similar as the observation-based estimate from surface ocean pCO2 observations (81 ± 7 Pg C) (Table 1)."

Reviewer 2 and this review provide conflicting advice with respect to the focus of this study. Reviewer 2 is asking us to emphasize more the relevance of our results for the Global Carbon Budget. We follow the advice of reviewer 2, an expert in the field of hindcast simulations and ESMs. Accordingly, and to avoid misunderstandings, we now write in the manuscript:

"The here provided improved estimate of the size of the global ocean carbon sink may help to close the carbon budget imbalance since 1850 (Friedlingstein et al., 2022)"

and

"Moreover, biases in these quantities and corrections for the late starting date may well be the reason for offset between models and observations over the last 30 years (Hauck et al., 2020; Friedlingstein et al., 2022). Although the here identified constraints cannot correct for misrepresentation of the unforced decadal variability, such variability plays likely a minor role when averaging results over longer periods. Indeed, we find good agreement between our estimate and the observation-based estimate from the Global Carbon Budget 2021 for the period from 1990 to 2020. This agreement suggests that the hindcast models underestimate the ocean  $C_{ant}$  uptake. This underestimation is thus likely the explanation for the difference between models and observation-based product in the Global Carbon Budget (Friedlingstein et al., 2022). However, the output of the Global Carbon Budget hindcast models is not publicly available for evaluating possible data-model differences for the inter-frontal sea surface salinity, the AMOC, and the Revelle factor. "

**Detailed arguments:**

**Regarding Robustness:**

Emergent constraints essentially rely on the relationship between biases in the models and the biases that result from them with regard to a particular outcome – here the ocean uptake of CO2. While this is a well-tested method, its limits always need to be carefully evaluated. This is especially the case when an attempt is made to improve knowledge about a process for which a lot of information is already available, such as the past and present uptake of CO2 by the ocean.

We agree.

A fundamental underlying assumption in the method is that while individual models can be (and should be) biased, there is no common bias across all models that would lead to an overall bias set of models. This assumption is violated here. None of the employed ocean models is eddy-resolving – meaning that they all share similar biases with regard to a number of critically important processes. The role of eddies for determining global ocean circulation is well established, particularly with regard to the processes in the Southern Ocean, where the interplay between Ekman drift induced overturning circulation and eddydriven circulation is particularly important (see Marshall and Speer (2012) and Rintoul (2018)) for determining the structure and magnitude of the subduction of mode and intermediate waters, i.e., the important conduits for how anthropogenic CO2 is entering the thermocline of the Southern Ocean. This process is not well captured by most coarse-resolution models, as evidenced, e.g., by their poorly modeled distribution of salinity. Lachkar et al. (2007) showed the impact of resolution on the uptake of anthropogenic CO2, CFCs and  $\Delta$ 14C quite impressively, highlighting how it not only alters the global uptake, but also the processes and the locations of the uptake. Given this evidence, I have substantial concerns that the relationship established here is as robust as the authors make us believe. (note on the side: this would not be the first time an emergent constraint falls apart once additional processes are taken into account).

**This comment was already made above. Please see our previous answer.**

I think also a bit more critical thinking would do this study well. One needs to recall that in the end, emergent constraints can only emerge from a model suite if at least some of the models are flawed. In addition, emergent constraints study often just emphasize the variables that work. They rarely state (also not in the case of this study) of all the variables that did not work. For example, it turns out that interfrontal salinity in the Southern Ocean ends up to be the most important constraint. But why not interfrontal density, which is actually dynamically the more important variable? And why not winds, and why not winter mixed-layer depths and why not many other variables that are clearly relevant for the determining the anthropogenic CO2 uptake in the Southern Ocean? The lack of consideration of the fact that these emergent constraints emerge from a substantial amount of trial and error approach also tends to lead to overconfidence.

**All authors find the wording of this comment (i.e., a bit more ciritical thinking) irritating.**

Taken the comment at face value, all scientists publishing a best estimate and uncertainty ranges are overconfident and uncritical as all models are "flawed" and by necessity only an approximation of reality.

The specific comment on salinity versus density is surprising as salinity variations are known to govern density variations in cold waters with limited temperature variations (see for example in Descriptive Physical Oceanography: An Introduction by Lynne D Talley). In Terhaar et al. (2021), we wrote:

"Across the CMIP6 and CMIP5 model ensembles the volume of ocean interior water ventilated by surface waters that lies between the PF and the STF, namely, SAMW and AAIW, increases with increasing sea surface density (r2 = 0.74; figs. S2 to S5). Sea surface density is, thus, a physically supported indicator of the formation rate of SAMW and AAIW (12, 41, 43, 44) and, in turn, of Cant uptake by the Southern Ocean (12, 41). The sea surface density variations in the cold Southern Ocean depends strongly on variations in surface salinity (r2 = 0.84; fig. S2A) (13, 42–45) and less on variations in surface temperature (r2 = 0.01; fig. S2B). [...]While the relationship between the volume of subducted SAMW and AAIW and the Southern Ocean Cant uptake might be more direct, we chose the sea surface salinity as the observable quantity because its observations are less uncertain. Sea surface salinity provides the best compromise between a good linear correlation and low observational uncertainties."

We added the following text to the revised manuscript:

"In addition, other potential predictors were tested, such as the magnitude and seasonal cycle of sea-ice extent, wind curl, and the mixed layer depth, and upwelling strength of circumpolar deep waters. All these variables are known to influence air-sea gas exchange, freshwater fluxes, and circulation and, in turn, salinity and  $C_{ant}$  uptake. However, none of these factors alone explains biases in the surface salinity and  $C_{ant}$  uptake in the Southern Ocean. Therefore, the sea surface salinity that emerges as a result of all these individual processes represents, so far, the best variable in terms of mechanistic explanation and observational uncertainty to bias-correct models for Southern Ocean  $C_{ant}$  uptake."

**Regarding data constraints.**

The authors compare their emergent constraints only with regard to the global uptake numbers with other data based constraints. But the proof of the pudding is the eating. Unless the authors can demonstrate that the constrained models are indeed doing better with regards to the observational constraints for the oceanic uptake of anthropogenic CO2, I have little confidence in their results. Of course, the observational constraints come with their own uncertainties, but there are a number of well established features in terms of basin and depth distributions that can be exploited (note e.g., that the Sabine et al. 2004 and the Gruber et al. 2019 estimates are statistically fully independent since they use a fundamentally different methodology). I also think that the ocean models should demonstrate their ability to represent the air-sea CO2 fluxes, since these are increasingly dominated by the anthropogenic CO2 flux components.

This comment was already made above. Please see our previous answer.

**Detailed comments:**

P5, line 116 "However, ... significantly smaller than the previously assumed flux of -5 Pg C (Gruber et al., 2019a),": Given that the ESMs employed here have their own climate variability, this comparison is fundamentally not tenable. The 5 Pg C could be related to anthropogenic climate change, but it could also be related to naturally occurring interannual to decadal climate variability. Thus the authors are comparing two different things here.

The estimate of  $C_{ant}$  uptake from 1994 to 2007 of 34 Pg C by Gruber et al. (2019) includes the steady-state  $C_{ant}$  uptake and the non-steady state  $C_{ant}$  uptake. However, it does not include the non-steady state flux of  $C_T$ . This flux includes natural fluxes of  $C_T$  across the air-sea interface due to climate change and climate variability. While the flux due to climate change leads to a cumulative flux over time, the flux due to climate variability should average out over several decades but may exist over a decade or two.

The ESMs quantify the effect of long-term climate change and  $CO_2$  increase. The ESM also simulated externally forced and internal climate variability. Using an ensemble of models and averaging over 14 years or longer typically removes unforced, internal variability, whereas forced variability (e.g., due to the voclanic eruptions such as Pinatubo or variations in emissions of  $CO_2$  or other radiative forcing agents) are preserved. The statement that ESMs have their own climate variability is thus partly misleading. There is little evidence for a relevant deviation of  $C_{ant}$  uptake due to climate variability during the period 1994 to 2007, both globally and in the Southern Ocean (Figure R.3). Uncertainties are simply too large.